# Molecular and phenotypic analysis of rodent models reveals conserved and species-specific modulators of human sarcopenia

Anastasiya Börsch[1,5], Daniel J. Ham [2,5], Nitish Mittal[1], Lionel A. Tintignac[3], Eugenia Migliavacca[4], Jérôme N. Feige [4], Markus A. Rüegg [2,6] & Mihaela Zavolan [1,6✉]

Sarcopenia, the age-related loss of skeletal muscle mass and function, affects 5–13% of individuals aged over 60 years. While rodents are widely-used model organisms, which aspects of sarcopenia are recapitulated in different animal models is unknown. Here we generated a time series of phenotypic measurements and RNA sequencing data in mouse gastrocnemius muscle and analyzed them alongside analogous data from rats and humans. We found that rodents recapitulate mitochondrial changes observed in human sarcopenia, while inflammatory responses are conserved at pathway but not gene level. Perturbations in the extracellular matrix are shared by rats, while mice recapitulate changes in RNA processing and autophagy. We inferred transcription regulators of early and late transcriptome changes, which could be targeted therapeutically. Our study demonstrates that phenotypic measurements, such as muscle mass, are better indicators of muscle health than chronological age and should be considered when analyzing aging-related molecular data.

---

[1] Biozentrum, University of Basel and Swiss Institute of Bioinformatics, Basel, Switzerland. [2] Biozentrum, University of Basel, Basel, Switzerland. [3] Department of Biomedicine, Pharmazentrum, University of Basel, Basel, Switzerland. [4] Nestlé Research, EPFL Innovation Park, Lausanne, Switzerland. [5] These authors contributed equally: Anastasiya Börsch, Daniel J. Ham. [6] These authors jointly supervised this work: Markus A. Rüegg, Mihaela Zavolan. ✉email: mihaela.zavolan@unibas.ch

Skeletal muscle is the largest human organ comprising up to 50% of the body weight. Primarily supporting movement, the muscle is also the main storage compartment for amino acids[1] and a major regulator of body metabolism. It responds to both load and systemic demands for energy or amino acids by modulating the rates of protein synthesis and breakdown[2]. Skeletal muscles are composed of long, cylindrical, and multinucleated skeletal muscle cells—muscle fibers—that are either slow-twitch (type I) or fast-twitch (type II), depending on whether they use predominantly aerobic respiration (oxidative phosphorylation) or anaerobic metabolism (glycolysis)[3,4].

Progressive loss of muscle mass is a striking feature of human aging, eventually leading to sarcopenia[5], a term that describes a reduction in muscle size and function to an extent that increases morbidity and mortality[6] and ultimately limits the ability of individuals to perform activities of daily living[7]. Morphological changes include reduced fiber number, increased heterogeneity in fiber size, selective loss of fast-twitch fibers, fiber type grouping, infiltration with non-muscle cells (e.g., adipocytes and connective tissue cells), and denervation[8]. About 5–13% of 60–70-year-old individuals are affected by sarcopenia[9], and the number increases to 11–50% for those aged 80 or above. Why some individuals are affected and others not is still unclear. Among hospitalized patients, sarcopenia has been linked to poor clinical outcome, and increases hospitalization costs by 58.5% and 34.0% for patients younger and older than 65 years, respectively[10]. Despite the high social and economic impact of sarcopenia, no medical treatment is yet available. Currently, the most successful interventions are lifestyle-based and include changes in caloric intake and food composition in combination with resistance and aerobic exercise[11].

Major challenges in studying sarcopenia are the long time over which the disease develops and the lack of noninvasive methods to estimate molecular changes in skeletal muscles during aging. This is where model organisms such as mice and rats, which exhibit morphological changes consistent with human sarcopenia after just 22–24 months of age, are extremely important[12–15]. In both rats and mice, muscle mass and function progressively decline with age[12–14], while mixed fiber types appear and a partial or complete muscle denervation occurs[12,16]. However, although numerous model strains have become available, including from physical[17] or genetic[18] perturbations, large-scale and phenotypically-backed molecular data across aging time lines are still lacking. Thus, the value of these models for understanding human sarcopenia is still debated[19–21].

Our study aims to provide an improved basis for investigating and modeling this disease. As sarcopenia is the age-related loss of muscle mass and function[22,23], we generated time series of both RNA-seq and extensive phenotypic measurements characterizing muscle aging in rodent model systems. Building on previous work in rats[12], we focused on the gastrocnemius muscle, expanding this data set and generating a comparable data set for mice. To identify conserved factors underlying sarcopenia, we analyzed our rodent data alongside human muscle samples from the Genotype-Tissue expression (GTEx) project[24]. We find that aging trajectories differ considerably among individuals within each species and changes in the muscle transcriptome more strongly reflect a muscle's morphological features than its chronological age. We identify genes and pathways whose dynamics reflect muscle state across species. We uncover previously unreported indicators of muscle impairment and show that despite differences in specific genes, most pathways respond similarly to aging in mouse, rat, and human muscle. We further infer the transcription factors (TFs) underlying these gene expression changes, which could be pursued in animal models for the development of new therapies. Finally, we integrate the muscle transcriptome profiles of mice

and rats analyzed in this study into "SarcoAtlas" (https://sarcoatlas.scicore.unibas.ch/), a resource which we have established to support further analyses of molecular mechanisms underlying sarcopenia[25].

## Results

**Emergence of the sarcopenic phenotype during mouse aging.** To follow the natural development of sarcopenia in male C57BL/6JRj mice, we comprehensively characterized changes in body composition, muscle mass and muscle function at 8, 14, 18, 22, 24, 26, and 28 months. Body mass (Fig. 1a) and composition (Fig. 1b, c) remained stable between 8 and 18 months. Body mass progressively declined after 18 months, reaching 15.2 ± 3.1% at 28 months compared to 8–18 months (Fig. 1a). EchoMRI measures of body composition revealed that progressive loss of fat mass (Fig. 1b) predominantly accounted for the loss of body mass, with lean mass (Fig. 1c) remaining relatively stable from 8 to 28 months. At the time of dissection, body fat was below 26% for all mice, consistent with a lean phenotype[26]. No overt signs of tumors were observed upon autopsy. In contrast to whole-body lean mass, the mass of all measured hind-limb muscles declined progressively with age reaching statistical significance by 22 months of age in all muscles, except soleus (26 months), compared to 8–18 months-old mice (Fig. 1d). By 28 months, the muscle mass decreased by 19.7% in the tibialis anterior and 29.6% in the quadriceps, compared to 8–18 months-old mice. Muscle loss outstripped body mass loss for all measured muscles, with the notable exception of the tibialis anterior, which was comparatively more resistant to age-related muscle loss, as previously observed in both humans[12] and mice[27]. Analysis of internal organs showed a maintenance (liver) or increase (heart) of other lean-tissue organs (Fig. 1e), explaining the overall maintenance of whole-body lean mass. Indeed, seminal vesicles, although not measured, were notably larger in older mice, as previously observed[27]. In contrast, and consistent with MRI measures of body composition, epididymal fat depositions were measurably lower in 22–28 months-old mice than 8–18 months-old mice, reaching a body mass normalized reduction of 81.6 ± 2.8% at 28 months.

As sarcopenia encompasses age-related loss of both muscle mass and function, we also measured functional parameters across these time points. All-limb grip strength decreased in parallel with muscle mass, reaching a 34.2 ± 3.4% loss in 28 months-old mice compared to 8–18 months-old mice (Fig. 1f). Next, we isolated the extensor digitorum longus (EDL) muscle to test muscle function directly (Fig. 1g–i). The EDL is a fast twitch, hindlimb muscle that can be isolated tendon-to-tendon and is thin enough for effective nutrient diffusion in an organ bath. The tetanic force was significantly lower at 22–28 months compared to 8–18 months above a stimulation frequency of 150 Hz (Fig. 1g, left). The reduction in tetanic force reached 28.9 ± 8.3% at 28 months at the stimulation frequency of 250 Hz. In response to repeated stimulations (200 Hz every 4 s for 4 min), the tetanic force in 28 months-old mice was significantly lower than in 8–18 months-old mice after 10 contractions, but all age groups converged thereafter reaching a similar plateau in tetanic force between 40 and 60 contractions (Fig. 1g, right). Analysis of EDL twitch properties showed a slowing of contraction speed in the oldest muscles with time-to-peak tension (Fig. 1h) and half relaxation time (Fig. 1i) significantly higher in 28-months-old mice compared to 8–18-months-old mice.

Together, these data show that while stable between 8 and 18 months, hindlimb muscle mass and function progressively decline in male C57BL/6JRj mice with the first minor decrements measurable from 22 months of age. Importantly, the temporal

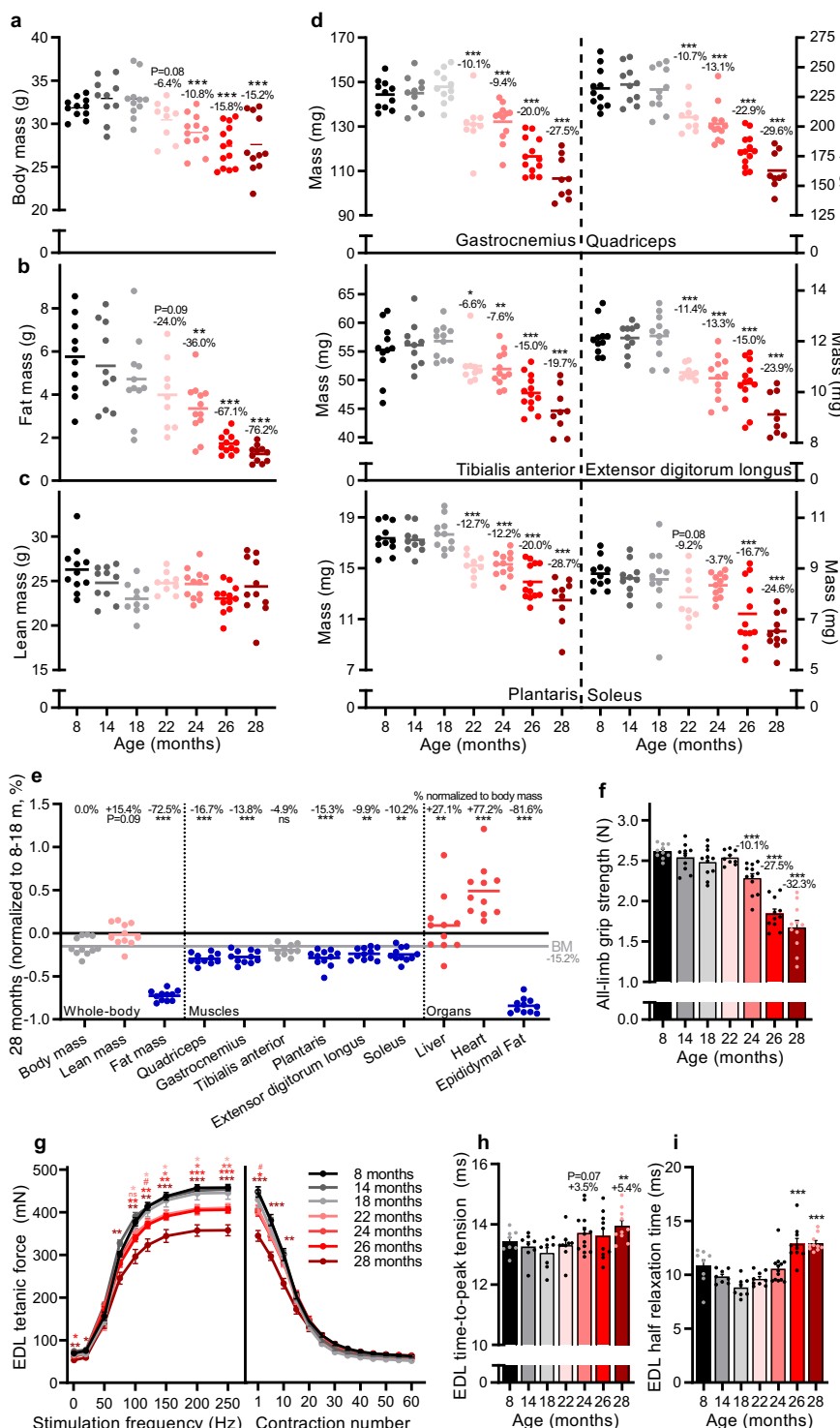

pattern and magnitude of reduction in skeletal muscle mass and function reach a level consistent with the clinical definition of sarcopenia[28] at 28 months of age.

**Skeletal muscle gene expression changes with age in mice, rats, and humans.** To identify conserved mechanisms of sarcopenia, we carried out a comparative analysis of our mouse molecular and physiological data, and of comparable data from rats and humans. We generated RNA-Seq data from the gastrocnemius muscles of the mice described above, collected at 8, 18, 22, 24, 26, and 28 months of age, corresponding to adult (8 and 18 months),

early sarcopenic (22 and 24 months) and sarcopenic (26 and 28 months) phases of muscle health status. For rats, we expanded a previous RNA-Seq time series from gastrocnemius muscle (8, 18, 24 months)[12] with two additional time points (20 and 22 months) and further included in the analysis available phenotypic measurements, the rat data thus covering the adult phase (8 months), the early sarcopenic phase (18 and 20 months) and the sarcopenic phase (22 and 24 months). Finally, for humans we used an RNA-Seq data set of gastrocnemius muscle from individuals aged between 22 and 70 years, which is publicly available from the Genotype-Tissue Expression (GTEx) project[24].

**Fig. 1 Muscle mass and function progressively decline in male C57BL/6JRj mice during aging. a** Body mass for 8, 14, 18, 22, 24, 26, and 28 months-old mouse groups. EchoMRI measurements of **b** fat and **c** whole-body lean mass. **d** Absolute muscle mass for quadriceps (QUAD), gastrocnemius (GAS), tibialis anterior (TA), plantaris (PLA), extensor digitorum longus (EDL) and soleus (SOL) averaged across both limbs. **e** Body, lean and fat mass as well as muscle tissue and organ mass in 28 months-old mice normalized to the mean of 8, 14, and 18 months-old groups. The mean percentage loss in mass relative to body mass is reported above each data set. The color scheme designates the direction of changes and significance: gray is not different ($p$-value > 0.10), red is increased ($p$-value < 0.05), blue is decreased ($p$-value < 0.05), light red is a trend for increased (0.05 < $p$-value < 0.10). **f** Recordings of all-limb grip strength. Isolated EDL muscle function parameters, including **g** force-frequency curve (left) and fatigue response to multiple stimulations (right); and twitch time-to-peak tension in (**h**) and half relaxation time in (**i**). Group numbers of biological replicates are: $n = 9$–13 in (**a-f**); $n = 7$–12 in (**g**) (left); $n = 5$–11 in (**g**) (right) and $n = 7$–12 in (**h-i**). For statistical comparisons 8, 14, and 18 months-old groups were pooled and compared with each of the other four groups. One-way ANOVAs with Dunnett's post hoc tests were used to compare between 8 and 18 months data and the other four groups. *, ** and *** denote a significant difference between groups of $p$-value < 0.05, $p$-value < 0.01 and $p$-value < 0.001, respectively. Trends (0.05 < $p$-value < 0.10) are denoted by # or the $p$-value specified. Colored asterisks refer to the group of comparison. All values were visualized as mean ± standard error of the mean (SE).

To reveal the structure in our data sets, we carried out principal component analysis (PCA) of transcript abundances estimated with the kallisto software[29]. In all species, sample coordinates on the first two principal components revealed a progressive transition from adult to sarcopenic phases, without clear age-based clustering (Fig. 2a–c). The heterogeneity in the molecular profile of muscles further increased with age (Fig. 2d–f, "Methods" section). Numerous genes exhibited changes in expression levels between old and young animals (Supplementary Fig. 1), though for the vast majority of genes, changes were smaller than 2-fold (Fig. 2g, h). The broad but nuanced changes in gene expression highlighted the complexity of muscle aging, and the inherent difficulty of discerning key drivers. Similar conclusions were recently reported for brain aging across species[30]. In humans, the number of genes reaching the threshold for statistical significance between young and old individuals was much smaller than for both rodent models (Fig. 2i), reflecting the additional heterogeneity in the human population, and possibly also the high variance in gene expression in old humans[31].

Altogether, these results indicate that the chronological age of individuals is not strictly linked to the health of their skeletal muscles, leading us to hypothesize that an analysis of the molecular data from the perspective of muscle health status, rather than age, is more likely to reveal factors specifically associated with sarcopenia.

**Transcriptomic profiles of skeletal muscles more closely reflect muscle health status than age.** As the coordinates of samples on the first principal component (PC1) spanned a substantial range, especially at high age (Fig. 2a–c), we sought to determine whether they better reflect muscle-specific properties, by correlating PC1 coordinates with various phenotypic measurements, e.g., muscle mass and grip strength of animals obtained from the corresponding animals. Muscle mass strongly correlated with PC1 coordinates in both mice and rats, stronger than the chronological age (Fig. 3a, Supplementary Figs. 2 and 3). This indicates that the standard approach of grouping samples by age[14,32] is likely to obscure important molecular mechanisms underlying sarcopenia.

For both mice and rats, the mass of other hindlimb muscles also correlated with PC1 of the gastrocnemius RNA-Seq data (Fig. 3a), suggesting that molecular data from one specific muscle also reflects the "health status" of other muscles in its vicinity.

PC1 coordinates of samples from aged animals with relatively high gastrocnemius mass were more similar to samples from young animals (Fig. 3b, c), and distinct from animals of the same age but with lower muscle mass. To further establish that PC1 was more specifically associated with the muscle loss than with the biological age, we split samples from 28 months-old mice into two groups "G1" and "G2" based on their PC1 coordinates and

muscle mass (Fig. 3b, dashed ellipses) and found that 1709 genes were differentially expressed between these groups at the significance threshold FDR < 0.01. The ontology terms enriched by these genes, i.e., "RNA splicing", "mitochondrion", and "oxidation-reduction process" (Supplementary Fig. 4), have known links to muscle aging[33,34]. These results indicate that animals of the same age can exhibit substantial differences in age-related molecular changes.

Age correlated poorly with the gene expression PC1 in humans (Pearson correlation coefficient of 0.27, Supplementary Fig. 5a). The higher complexity of these data was further highlighted by the relatively low percentage of variance (19%) in gene expression explained by PC1 compared to mouse (38%) and rat (32%; Fig. 2a–c). Sample collection may explain some of the variability between human individuals, since ischemic time and brain pH also show some correlation with PC1 of the muscle molecular data (Supplementary Fig. 5a). In spite of this difference, the distribution of samples on PC1 followed a similar pattern in humans as in other species, with higher between-subject variance in PC1 coordinates for individuals 46–70 years of age compared to those 45 years or younger (Supplementary Fig. 5b). Therefore, we analyzed the human data similarly to the data from rodent species, from the perspective of PC1. Nevertheless, to verify the assumption that PC1 of the human data reflects muscle health, further studies of human sarcopenia combining thorough phenotyping with molecular profiling of muscle samples are needed.

**Muscle aging involves largely similar molecular pathways in rodents and humans.** As the above results indicate that PC1 of the molecular data reflects changes in muscle health status, we analyzed the gene expression dynamics from the perspective of PC1[35,36], calculating the projections of gene expression vectors in mouse, rat and human on their corresponding PC1 (Supplementary Fig. 6a). As expected, these projections were normally distributed for all considered species (Supplementary Fig. 6b) and thus, for comparing across species we standardized the projection values by calculating corresponding z-scores (Supplementary Fig. 6c). For further discussion, we note that expression of genes with positive projection z-scores increased whereas that of genes with negative projection z-scores decreased during sarcopenic progression.

To compare the dynamics of gene expression in relation to muscle functionality between species, we correlated the projection z-scores for individual genes between species (Fig. 4a–c). We obtained relatively low inter-species Pearson correlation coefficients, with the highest correlation value for human and mouse ($r = 0.35$) and the lowest for mouse and rat ($r = 0.10$). The reasons behind these relationships remain unclear.

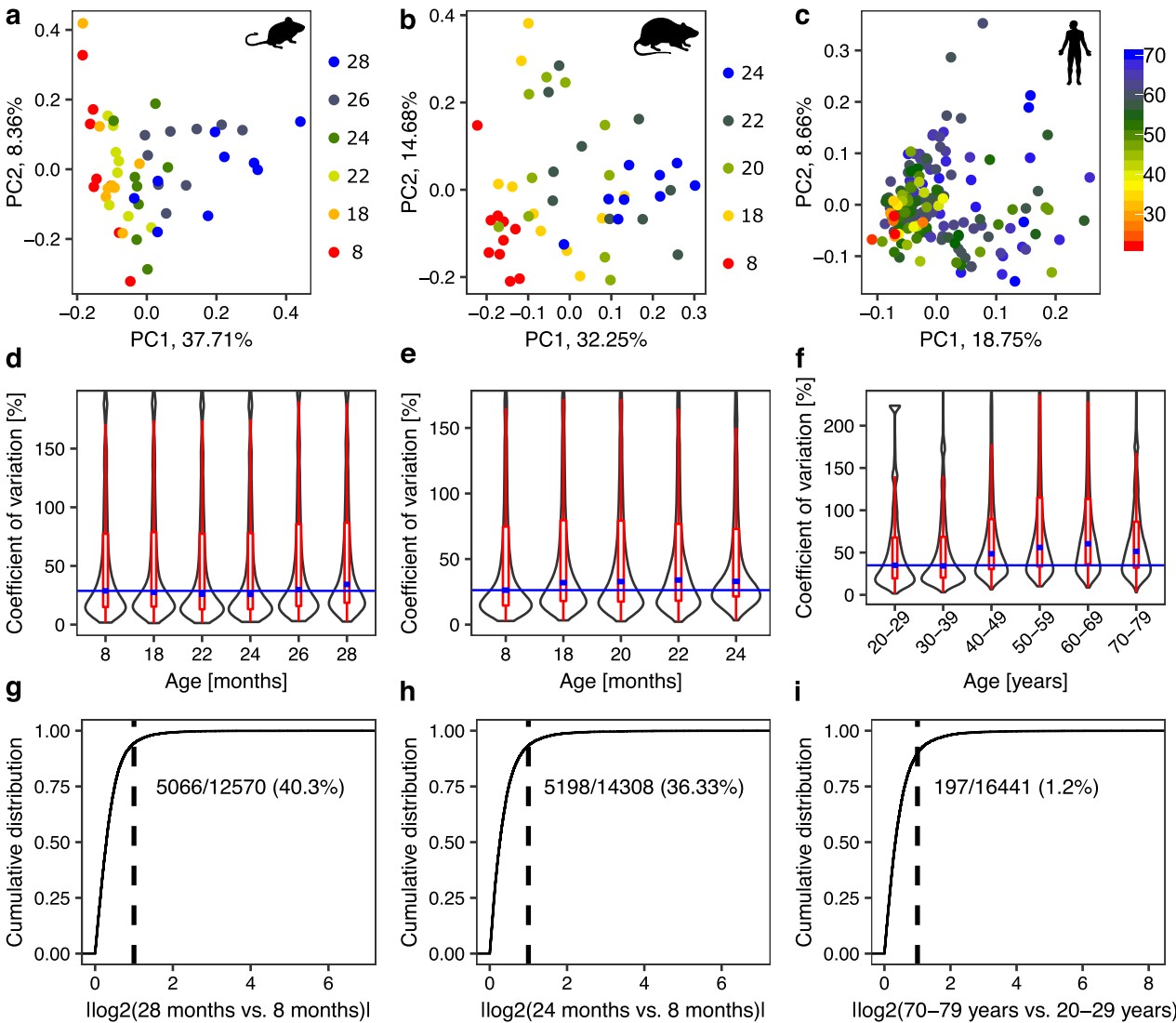

**Fig. 2 Summary of gene expression changes in the gastrocnemius muscle during aging across species. a–c** Principal component analysis (PCA) of transcript abundances during muscle aging in mice in (**a**), rats in (**b**) and humans in (**c**). Each dot corresponds to one sample with colors indicating organism age. The numbers associated with PCs indicate the fraction of the variance in gene expression across samples along the corresponding PC. **d–f** Distribution of the coefficients of variation (CVs) of individual genes per age/age group for mice in (**d**), rats in (**e**) and humans in (**f**). The higher the CV, the more variable the expression of the gene across replicates. Thin blue lines are baselines indicating the median CV for the youngest age/age group. Median values of CVs for other ages/age groups (thick blue lines within violin plots) are mostly located over the baseline, especially for older ages/age groups, indicating higher heterogeneity across replicates for these ages/age groups in comparison to the youngest one. Limits for y-axis were set to include both the 25th and 75th percentiles of data points and up to 1.5 times the interquartile range in both directions from percentiles (red). **g–i** Cumulative distribution of absolute log2-fold changes in gene expression between the youngest and oldest age groups of mice in (**g**), rats in (**h**) and humans in (**i**). Differential expression analysis was performed with the EdgeR tool[31]. Dashed lines designate the position of the log2-fold change of 1, numbers correspond to the fraction of genes with significantly different expression among all genes in the group of oldest compared to the group of youngest individuals (FDR < 0.01). Group numbers for age groups of biological replicates are: $n = 8$–9 for mouse, $n = 9$–10 for rat and $n = 5$–79 for human.

To identify KEGG pathways that are associated with loss of gastrocnemius functionality during aging, we performed gene set enrichment analysis (GSEA)[37,38]. Figure 4d explains the concept of the analysis using two representative pathways "Oxidative phosphorylation" (blue) and the "Jak-STAT signaling pathway" (red) in the mouse data set. Briefly, for each species, we ranked genes by their projection z-scores (Fig. 4d, bottom), and then we located the genes corresponding to a specific annotated pathway in the ranked gene list (blue and red vertical bars). Based on the distribution of genes in this list (Fig. 4d, top), the enrichment score (ES), normalized enrichment score (NES), and the significance rate of the enrichment (FDR) were calculated. They reflect the degree to which genes in the pathway are overrepresented at the top or bottom of the ranked gene list. A positive or negative ES (or NES) indicates that the pathway is enriched with genes whose expression increases (e.g., "Jak-STAT signaling pathway") or decreases (e.g., "Oxidative phosphoryla-tion"), respectively, during muscle aging. We retained for further analysis pathways for which the FDR was less than 0.01 in at least one species.

Applying this analysis to all data sets and all pathways we found that while the involved genes differed between species, age-related gene expression changes came from similar pathways in all species (Fig. 4e). Pathways with increasing gene expression during muscle aging were related to immunity and inflammation ("NF-kB signaling pathway", "Jak-STAT signaling pathway",

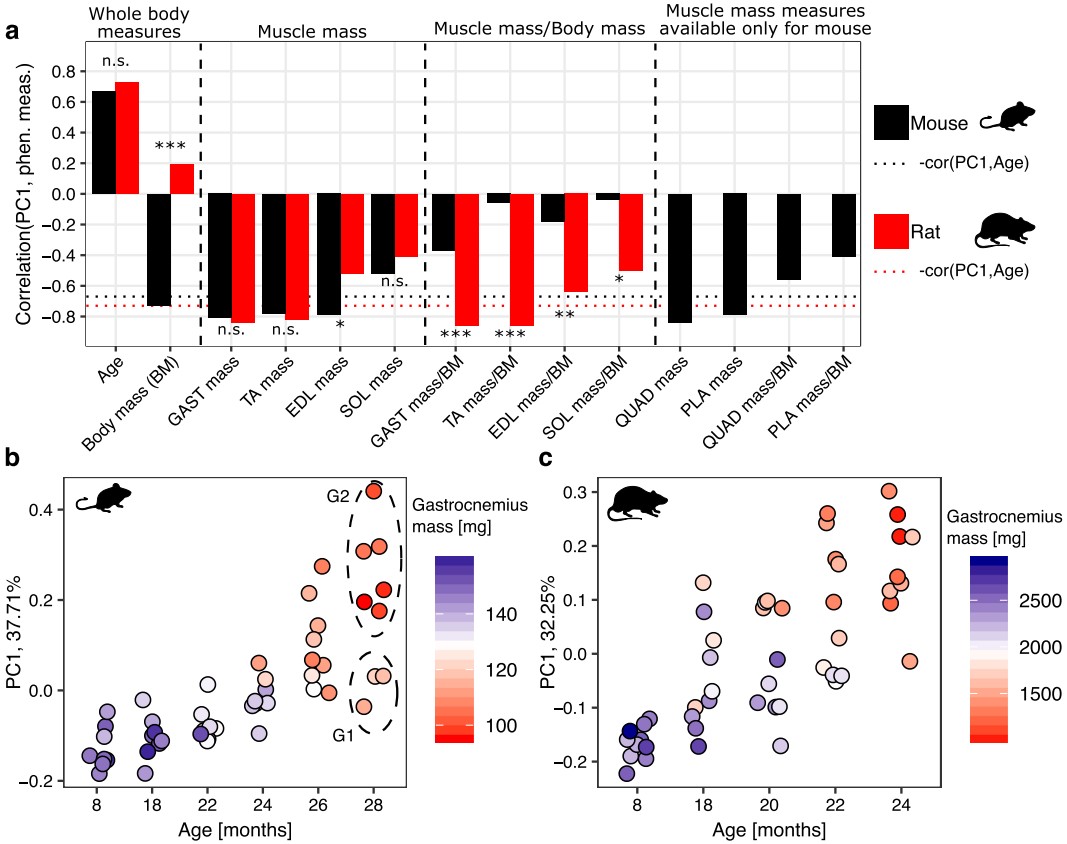

**Fig. 3 Correlation between PC1 and phenotypic measurements in rodents. a** Values of the Pearson correlation coefficient between PC1 and phenotypic measurements for mice (black) and rats (red) during aging. Black and red dotted horizontal lines are baselines representing the negative value of the correlation coefficient of PC1 with age for mouse and rat, respectively. Fisher's test[92] was used to compare Pearson correlation coefficients obtained for the same measure for mouse and rat, *p-value < 0.05, **p-value < 0.01, ***p-value < 0.001, "n.s." not significant (p-value ≥ 0.05). **b, c** PC1 coordinates of mouse and rat RNA-Seq data sets, respectively, grouped by age and colored by the gastrocnemius mass. With dashed ellipses we marked two groups of 28 months-old mouse replicates ("G1" and "G2") used for subsequent differential expression analysis. BM body mass, GAST gastrocnemius, TA tibialis anterior, EDL extensor digitorum longus, SOL soleus, QUAD quadriceps, PLA plantaris.

"TNF signaling pathway", and "Cytokine-cytokine receptor interaction"), as well as cell growth and proliferation ("p53 signaling pathway", "Cell cycle", "Cellular senescence", and "PI3K-Akt signaling pathway"). In contrast, genes from metabolic pathways ("Oxidative phosphorylation", "Citrate cycle", and "Glycolysis") decreased their expression during muscle aging. Some pathways were shared between two species only. For example, RNA homeostasis ("RNA transport" and "Spliceosome"), translation ("Ribosome biogenesis"), and turnover of proteins and organelles ("Autophagy" and "Mitophagy") were shared between mice and humans only and were enriched with genes that increased their expression during muscle aging. Pathways related to the extracellular matrix (ECM, "ECM-receptor interaction", and "Focal adhesion") and metabolism ("Propanoate metabolism") were shared only between human and rat and enriched with genes increasing their expression during muscle aging. Finally, "Fructose and mannose metabolism" and "Pentose phosphate metabolism" were shared only between rodents and enriched with genes decreasing their expression during muscle aging. Altogether, these results show that although many changes associated with muscle aging are shared between human and rodents, the conservation is stronger at the level of pathways than at the level of individual genes.

A central question in aging studies is to identify the key underlying events that drive aging. Thus, to gain insight into the sequence of age-related changes in muscle, we asked at which time points genes in the conserved pathways exhibit the most

pronounced changes. To answer this question, we identified "leading-edge" genes[37] for each organism and each pathway that was significantly enriched in all species (GSEA, FDR < 0.1). Leading-edge genes account for the pathway's enrichment signal (Fig. 4d, genes surrounded by boxes). For each of these genes, we calculated the mean expression across replicates within the same age group in rodents and among individuals of 20–29, 30–39, 40–49, 50–59, 60–69, and 70–79 years in humans. For each pair of consecutive time points we then calculated the median value of the slopes of gene expression changes across all leading-edge genes and depicted them as a heatmap (Fig. 5a and Supplementary Fig. 7). We found that in mouse, the most pronounced changes occur during the late sarcopenic phase (26–28 months). In contrast, the most pronounced changes occur at the beginning of the sarcopenic phase (22 months) in rats. Surprisingly, in humans there were two major waves of gene expression changes: an early wave with pathway-specific timing and a late wave (from 60–69 to 70–79 years) that occurred in all pathways. Pathways related to oxidoreductase activity and energy metabolism were the first to change from 20–29 years to 30–39 years, whereas pathways related to inflammation and proliferation changed from 30–39 years to 40–49 years. This analysis reveals that remodeling of energy metabolism precedes the onset of inflammation and proliferation in human muscle.

Finally, we checked if leading-edge genes of commonly regulated pathways were shared among species and found that the extent of sharing was pathway specific (Fig. 5b). For example,

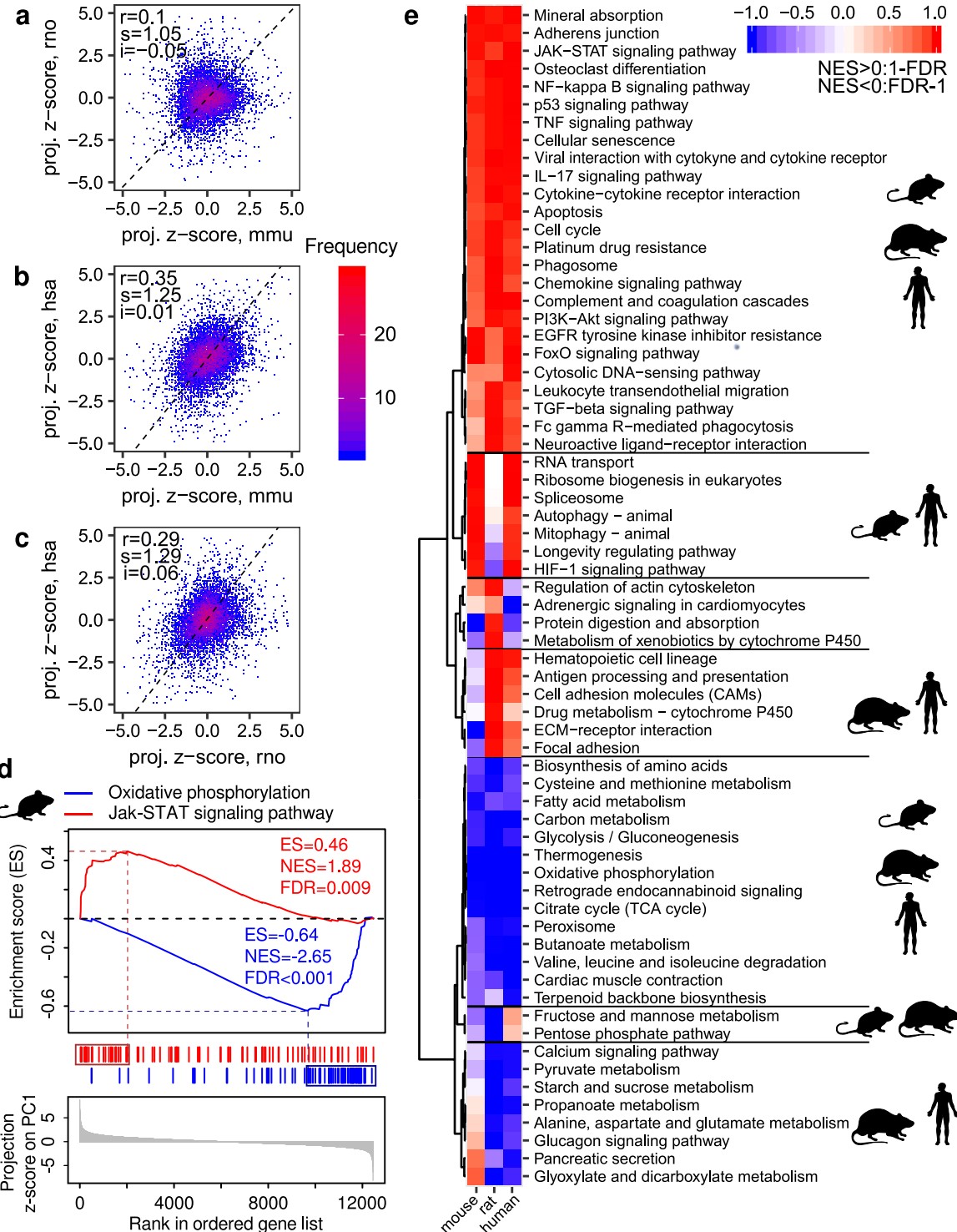

**Fig. 4 Changes in gene expression and pathway activities during gastrocnemius aging across species. a–c** Correlation between the standardized PC1 projections for individual genes (projection z-scores) across species. "mmu", "rno" and "hsa" designate "*Mus musculus*" (mouse), "*Rattus norvegicus*" (rat) and "*Homo sapiens*" (human), respectively. "r" indicates the value of the Pearson correlation coefficient. Black dashed lines correspond to directions of the highest variance for comparisons, with the slope "s" and intercept "i". **d** Schema of the gene set enrichment analysis (GSEA) for the KEGG pathways "Oxidative phosphorylation" and "Jak-STAT signaling pathway" for the mouse gastrocnemius time course RNA-Seq data. ES enrichment score, NES normalized enrichment score. **e** Heatmap summarizing the enrichment of KEGG pathways among genes ranked by projection z-scores for mouse, rat, and human, respectively. A pathway was included in the heatmap if it was significantly enriched in at least one organism with the significance threshold FDR < 0.01. Hierarchical clustering revealed pathways with a similar response during muscle aging in two or more species.

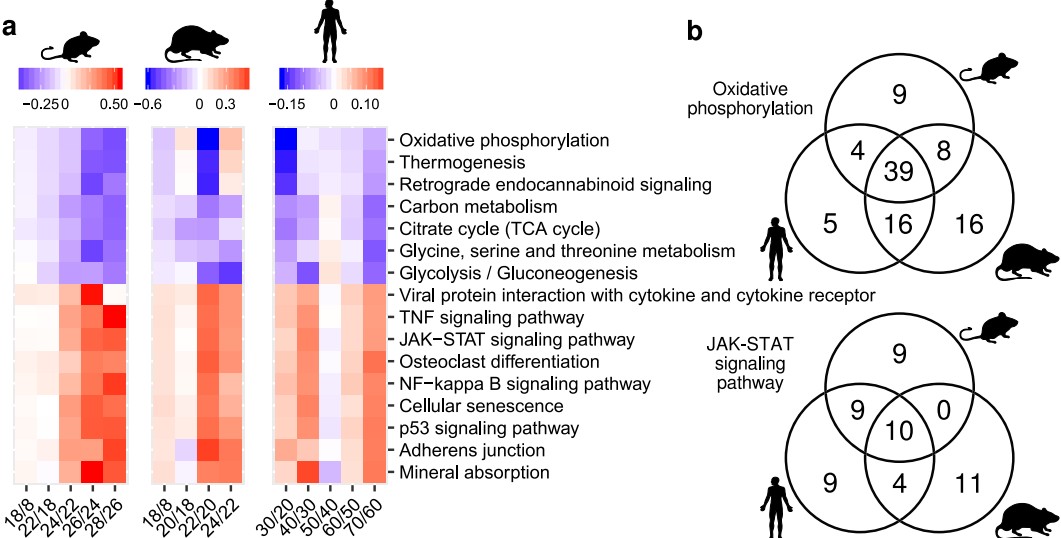

**Fig. 5 The timing of changes in core age-related pathways. a** Slopes of changes in the expression of genes from KEGG pathways that were significantly enriched for all considered species in GSEA (FDR < 0.1). The mean expression in replicates of the same age for each gene from the leading edge was calculated (see further description in the text). For humans, the mean expression was calculated for replicates from the age groups 20–29, 30–39, 40–49, 50–59, 60–69, and 70–79 years. The slopes defined by mean gene expression changes in neighboring time points (or age groups) were used to calculate median values across genes from the leading edge, which are visualized (Supplementary Fig. 7). **b** Venn diagrams of genes from the leading edge of two representative pathways across species.

in the "Oxidative phosphorylation" pathway the majority of leading-edge genes were common across species, indicating that changes in energy metabolism that occur upon age-related loss of muscle mass and function are evolutionary conserved. In comparison, leading-edge genes of pathways related to immune response and inflammation such as the "Jak-STAT signaling pathway" were mostly species-specific, suggesting that the regulation of these processes during muscle aging may differ between species.

**Conserved transcription regulators associated with age-related muscle loss.** The conserved, coordinated response of multiple pathways in spite of moderate correlations in the response of individual genes suggests the action of upstream regulators, transcriptional (transcription factors, TFs) or post-transcriptional (e.g., miRNAs). Thus, we used the ISMARA tool[39] to first infer which TFs and miRNAs were most active in our samples and then the highest confidence targets of these regulators. Then, for each pathway commonly regulated during aging across species (Fig. 5a), we identified the 10 regulators that had most genes from the leading edge among their top 300 ISMARA-predicted targets. Strikingly, a group of conserved TFs (estrogen related receptor proteins (ESRRs), Nuclear Receptor Subfamily 5 Group A Member 2 (NR5A2), peroxisome proliferator activated receptor alpha (PPARA), and ying-yang proteins (YY1/2), Fig. 6a and c–k, Supplementary Figs. 8 and 9) were predicted to regulate metabolism and mitochondrial function-related genes, whose expression decreased during muscle aging.

In contrast, the genes whose expression increased during muscle aging did not share a common regulator across all species (Fig. 6b). Rather, NFKB was predicted to regulate some of these genes in humans and rats, the Ets family TF SPI1 in mouse and rat, while MLX interacting protein like (MLXIPL), a TF from the Myc/Max/Mad superfamily, was predicted to regulate inflammation-related genes in humans and mice. Interestingly, a recent study reported that the Myc family TFs are induced by systemic aging in both mouse and human myogenic progenitors[40].

To partially validate our predictions, we checked the RNA levels of TFs commonly regulated across species in our data sets. RNA expression of *ESRRA* was significantly downregulated during muscle aging in all species, while *PPARA* decreased significantly in rat and human aging (Supplementary Fig. 10). RNA levels of other TFs whose motifs exhibited age-related changes were either unchanged or too low for statistical testing. We further analyzed protein expression of the TFs ERRα, PPARα, and YY1, whose activity was predicted to decrease with age in all species, by western blotting from mouse muscle tissue (Fig. 7a, Supplementary Fig. 11). In line with our predictions, the relative abundance of ERRα, PPARα, and YY1 was lower in 28 than 8 months-old mouse gastrocnemius muscle (Fig. 7a, b). Since ERRα exerts its activity in the nucleus, we further quantified the prevalence of nuclei strongly positive for ERRα staining (ERRα+; >50% of nuclear area with positive ERRα staining) in 8 and 28 months-old tibialis anterior muscle cross sections (Fig. 7c). We found ~80 ERRα+ nuclei per 100 muscle fibers in 8-months-old muscle, and 4-fold lower numbers in the 28-months-old muscle (Fig. 7d). Together, these data support our computational predictions of TF activity in aging muscles, and indicate that conserved changes in metabolism and mitochondrial function are driven by a set of conserved TFs in all three species.

## Discussion
Age being now seen as a risk factor for many diseases[41], targeting the aging process per se is expected to improve life quality and reduce the economic burden of hospitalization in elderly populations[42]. As with many human diseases, aging is studied in model organisms, preeminently in rodents[43]. However, while many physiological mechanisms are conserved between humans and rodents, whether this holds for the long and complex aging process is not well understood[44]. Even among rodents, aging does not appear to follow an identical timeline, as life expectancy varies not only across species, but also across strains[45,46]. In this study, we have indeed observed that mice maintained their muscle function well until ~24 months of age, while sarcopenic symptomatology occurred earlier in rats, around the age of

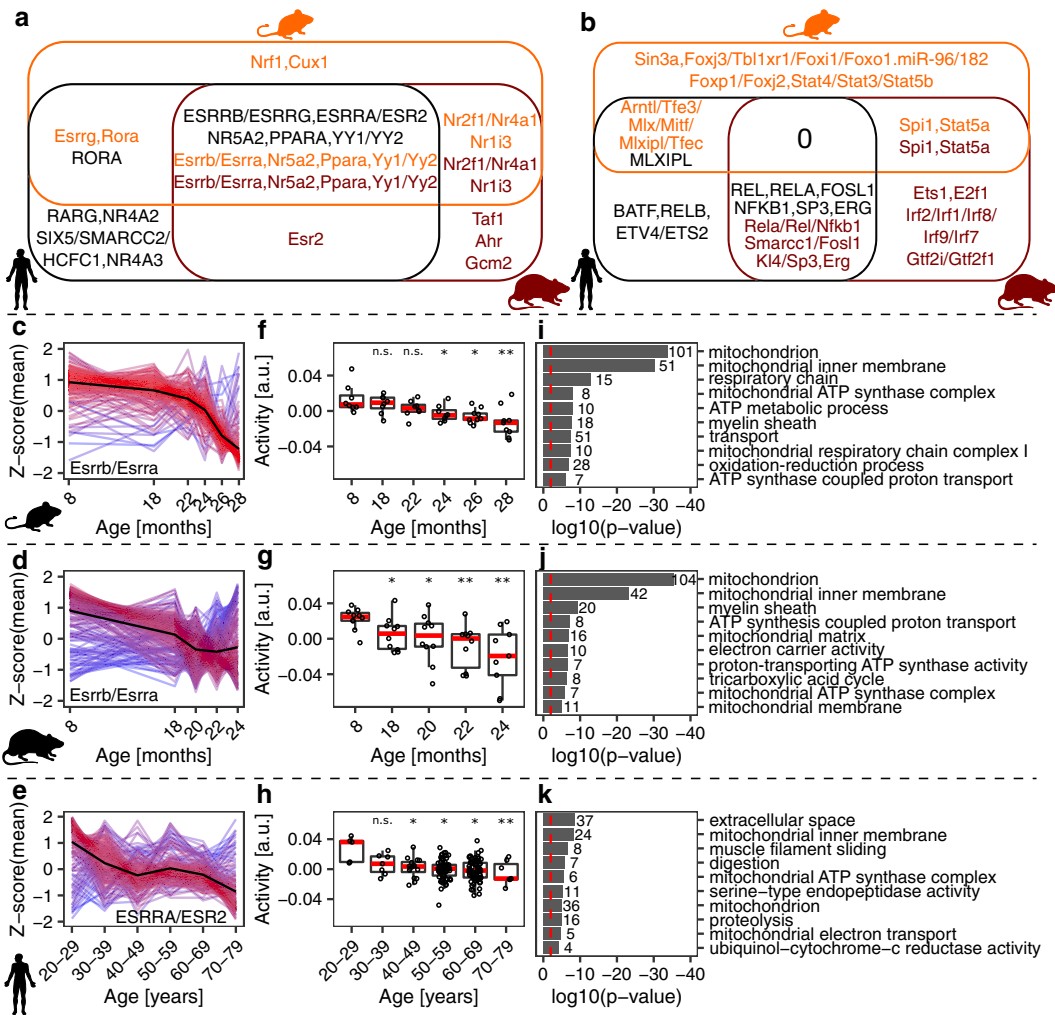

**Fig. 6 ISMARA-inferred[39] activity of transcription factors (TFs) and miRNAs during muscle aging. a** Venn diagram of motifs associated with TFs and miRNAs whose targets were downregulated during muscle aging. **b** Venn diagram of motifs associated with TFs and miRNAs whose targets were upregulated during muscle aging. Names of TFs and miRNAs are color-coded with respect to species: orange—mouse, brown—rat, black—human. **c–e** The normalized expression (z-scores of mean log2(TPMs)) of the top 300 target genes of motifs Esrrb/Esrra, Esrrb/Essra, and ESRRA/ESR2 in mouse, rat, and human, respectively. The mean value per age (or age group) across genes is indicated by the black (reference) line. Gene expression time course lines were colored by the distance from the reference line: red- close to the reference line, blue- far from the reference line. **f–h** Activity of motifs associated with TFs Esrrb/Esrra, Esrrb/Essra and ESRRA/ESR2 in mouse, rat, and human, respectively, predicted by ISMARA. *, ** and *** denote a significant difference based on two-sided Student's *t*-test between the youngest age/age group and all other ages/age groups with *p*-value < 0.05, *p*-value < 0.01 and *p*-value < 0.001, respectively; "n.s." not significant (*p*-value ≥ 0.05). **i–k** The 10 most enriched GO terms for the top 300 target genes of the TFs, i.e., genes depicted in (**c–e**). GO analysis was performed in DAVID[87]. Red dashed lines indicate the significance threshold (*p*-value < 0.01). The numbers next to the bars denote how many genes were attributed to an enriched GO term.

20 months. It is known that the Wistar rat strain used in this study develops peripheral neuropathy during aging[12]. This leads to muscle loss specifically in the hind limbs, where the studied gastrocnemius muscle is located. As a consequence of the earlier emergence of sarcopenia these animals do not experience as extensive age-related weight loss as mice, and lack features that develop only late in life. Therefore, muscle mass and body mass-normalized muscle mass were similarly well-correlated with PC1 of molecular data in rats (Fig. 3a). In mice, normalization by body mass strongly reduced the correlation with PC1, underscoring the importance of considering phenotypic parameters in analyzing the molecular data, and the fact that PC1 reflects the muscles health status.

An additional complication is that the lifespan represented in various data sets may differ for different species. Indeed, our mouse data corresponds to a much higher age compared to the

data for rats and humans, as 24 months of age in rats is thought to correspond to 60 years of age in humans[47], and 20 months of age in mice[48]. This could explain the larger number of genes found as differentially expressed between "young" and "old" mice compared to humans (Supplementary Fig. 1).

The inter-individual variability of aging rates within the same species/strain, underlying the observed increase in variance of various parameters with age[30], is a challenge for aging research, where samples are typically stratified by chronological age[13,14,32]. The human population is particularly heterogeneous, the median value of the coefficient of variation of expression levels among replicates from the same age group being much bigger for humans (34% and 60%) than for rodents (26% and 34%) (Fig. 2d–f, thick blue lines within violin plots). We found that the biological age of the animal is a poorer indicator of muscle functionality than the muscle mass. Taking into account

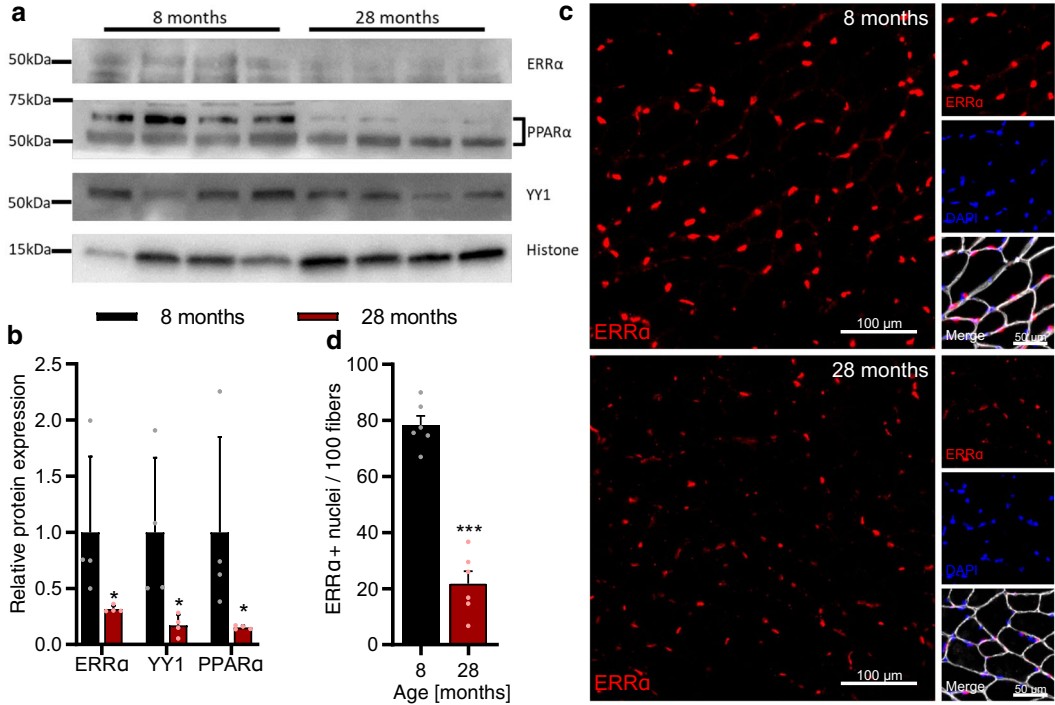

**Fig. 7 Validation of predictions for the ISMARA-inferred[39] activity of transcription factors (TFs) during muscle aging. a** Representative western blot analysis of the abundance of TFs ERRα, YY1, and PPARα in the gastrocnemius muscle (tissue lysate) of 8 and 28-months-old mice, respectively. **b** Quantification of western blots showing the relative abundance of TFs ERRα, YY1, and PPARα normalized to the nuclear protein histone H3, respectively. *Denotes a significant difference based on two-sided Mann–Whitney U test between 8 and 28 months-old mice with p-value < 0.05. **c** Representative images of tibialis anterior cross sections of 8 and 28-months-old mice with magnification stained for ERRα (red), Laminin α2 (white), and DAPI (blue). **d** Quantification of the percentage of ERRα-positive nuclei in tibialis anterior fibers of 8 and 28-months-old mice, respectively. ***Denotes a significant difference based on two-sided Student's t-test between 8 and 28 months-old mice with p-value < 0.001. All values were visualized as mean ± standard error of the mean (SE).

parameters of muscle health status reveals much more clearly the conservation of molecular processes among species (Fig. 4e).

Since fiber composition varies among muscles and age groups[12], we here focused on the same hind-limb muscle, gastrocnemius, for which data are available in all three species. Core pathways with established function in aging (inflammation, protein synthesis, cell division, and DNA damage) emerged also from our study of this muscle[49], although changes in individual genes were small and poorly correlated among species. Some pathways underwent age-related changes in only two of the three species. Common to mice and humans were pathways linked to the mechanistic target of rapamycin complex-1 (mTORC1), which controls the balance between protein synthesis and degradation in literally all tissues. mTORC1 activity increases during skeletal muscle aging[50–52]; its long-term inhibition with rapamycin extended the mouse lifespan[53], and positively impacted the skeletal muscle of mice[25]. Our study supports the notion that the mouse is a good model for studying the dynamics of mTORC1 signaling during aging. In contrast, pathways related to the extracellular matrix (ECM) underwent similar changes in human and rat. ECM provides a structural support and force transmission system for skeletal muscles[54], but also serves as a reservoir for growth factors and metabolites[55,56]. Numerous studies reported a decrease in expression of ECM components during muscle aging in animal models[14,57], whereas in humans, the concentrations of ECM components were reported to either increase[58] or remain unchanged[59]. Our study suggests that the rat is a good model organism for studying the poorly understood role of ECM during muscle aging[60].

A burning question in the aging field concerns the earliest signs of the process[61]. We found that in rodents, molecular changes occurred relatively late during the sarcopenic phase, while in humans there were two waves of change, metabolic remodeling in the young adult preceding changes in cellularity and inflammation. This finding is in line with a recent study of the plasma proteome across human lifespan, where the alterations in distinct biological pathways also occurred in waves in the fourth, seventh, and eighth decades of life[62]. These metabolic changes, which are more similar across species, have an early onset in humans suggesting that anti-aging interventions should start relatively early.

To identify possible targets of such interventions, we have inferred regulators that underlie changes in gene expression[39]. We associated transcription factors (TFs) such as estrogen-related receptors (ESRRs) ESRRA, ESRRB, and ESRRG, Peroxisome proliferator activated receptor Alpha (PPARA), Nuclear Receptor Subfamily 5 Group A Member 2 (NR5A2), and ying-yang proteins (YY1/2) with gene expression changes in mitochondria-related processes in all species (Figs. 6 and 7, Supplementary Figs. 8 and 9). ESRRs regulate mitochondrial biogenesis in multiple tissues. They are responsible for exercise tolerance and muscle fitness[63,64], and ESRRA deficiency in skeletal muscle impairs regeneration in response to injury[65]. PPARA modulates mitochondrial fatty acid oxidation in various tissues including skeletal muscles[66]. PPARA-null mice demonstrate profound changes in lipid metabolism during aging[67], and PPARA over-expression in skeletal muscles protects mice from diet-induced obesity[68]. Both ESRRs and PPARA are downstream targets of Peroxisome proliferator-activated receptor gamma coactivator 1-

alpha (PGC-1α)[69,70], a TF which is also downregulated during muscle aging in rodents and humans[71,72]. PGC-1α protects skeletal muscles from proteolysis, oxidative damage, inflammation, uncontrolled autophagy, and apoptosis[73,74], and low PGC-1α/ESRRA signaling and downregulation of oxidative phosphorylation and mitochondrial proteostasis were observed in human individuals with sarcopenia[34]. In our data, *PPARGC1A* downregulation was only significant in the aging rat (Supplementary Fig. 12). Altogether, these data indicate that the activity of ESRRs and PPARA, and of their common regulator PGC-1α should be monitored during human aging to identify individuals at risk of sarcopenia.

Another mitochondrial regulator, NR5A2/LRH-1[75], stimulates glucose metabolism in muscle cells[76]. This process is downregulated during muscle aging in all species (Fig. 4e), consistent with the reduced activity of NR5A2 (Supplementary Fig. 8d–f). The precise impact of YY1/2 TFs in muscle aging remains to be studied, as they exert broad control over many cellular processes, including proliferation and apoptotic signaling[77], both regulated during muscle aging in all species (Fig. 4e). Altogether, our analysis suggests that physiological and pharmacological interventions to increase the expression of these TFs in skeletal muscles during aging could improve mitochondrial biogenesis and provide a new means to counteract the development of sarcopenia.

Somewhat surprisingly, we found much less conservation of the regulatory control of inflammation, immune processes, and the cell cycle, whose associated genes increased the expression during muscle aging (Fig. 6b). However, MLX interacting protein like (MLXIP), nuclear factor kappa-light-chain-enhancer of activated B cells 1 (NFKB1) and the Ets family TF SPI1 had similar inferred activity in mouse and human, rat and human and mouse and rat, respectively. These differences between the model organisms used in the study of human sarcopenia may explain some debates in the field[19–21].

An outstanding question concerns differences between sexes in the emergence and development of sarcopenia. Previous studies suggested that the rate of absolute muscle loss during aging is sex-specific[78], contributing risk factors being malnutrition in females and higher serum myostatin in males[79]. Although due to data availability in mice and rats our study focused on males, we attempted to address the issue of sex-specific changes in humans, for which molecular data were available in the GTEx database (Supplementary Fig. 13). We found strongly correlated, age-related gene expression changes ($r = 0.91$) in males and females and a similar pattern of pathway activity changes. Thus, our study indicates that at the molecular level, aging-related changes in human gastrocnemius muscle are very similar between females and males.

In conclusion, our results demonstrate that rodent models recapitulate molecular changes observed in human sarcopenia, but the timing of aging-related changes is gradual in rodents and with two distinct waves in humans. Furthermore, the conservation is stronger at the level of pathways than at the level of individual genes. The RNA-seq data from this study can be found in "SarcoAtlas" (https://sarcoatlas.scicore.unibas.ch/)[25], an interactive application for exploring gene expression changes in multiple skeletal muscles during aging and in response to a genetic intervention and/or drug treatment.

## Methods

**Animal care**. Male, C57BL/6JRj mice were purchased from the aging colony at Janvier Labs (Le Genest-Saint-Isle, France).

Male C57BL/6JRj mice were kept on a 12 h light-dark cycle (6 am to 6 pm). For 8, 14, and 18 months time points mice were acclimatized to an AIN-93M diet (KLIBA NAFAG) for at least 2 months prior to endpoint measures. Mice used for

22, 24, 26, and 28 months time points received an AIN-93M diet from 18 to 19 months of age until endpoint experiments. Mice displaying overt signs of illness or tumors at the time of dissection were excluded from the study.

All mouse related procedures were performed in accordance with Swiss regulations for animal experimentation and approved by the veterinary commission of the Canton Basel-Stadt.

**Body composition analysis**. Body composition, including fat and lean mass, was analyzed using an EchoMRI-100 (EchoMRI Medical Systems) in restrained, conscious mice.

**Grip strength**. All-limb grip strength was measured by placing mice on a small grid attached to a force meter (Columbus Instruments). Once the mouse gripped firmly onto the grid with all four paws, it was gently pulled horizontally at a consistent speed until the grasp was broken. Performance was measured as the median of 3–5 trials with at least 10 min rest between tests. Trials where the mouse actively pulled on the grid while the test was underway were discarded. The same researcher performed all grip strength measurements at a similar time of day.

**In vitro muscle force**. The in vitro muscle force measurements of the extensor digitorum longus (EDL) were described previously[80]. Briefly, muscles were carefully excised and mounted on the 1200A Isolated Muscle System (Aurora Scientific, Aurora, ON, Canada) in an organ bath containing 60 mL of Ringer solution (137 mM NaCl, 24 mM NaHCO$_3$, 11 mM glucose, 5 mM KCl, 2 mM CaCl$_2$, 1 mM MgSO$_4$, 1 mM NaH$_2$PO$_4$) that was gassed with 95% O$_2$; 5% CO$_2$ at 30 °C. After defining the optimal length, muscles were stimulated with 15 V pulses. Muscle force was recorded in response to 500 ms pulses at 1–250 Hz. Muscle fatigability was assessed using 200 Hz stimulations every 4 s for 4 min (i.e., 60 contractions).

**RNA extraction**. Snap frozen gastrocnemius muscles of male C57BL/6JRj mice were pulverized and lysed in RLT buffer (Qiagen) and then treated with proteinase K (Qiagen). DNAse treatment and RNA extraction was performed with an automated iColumn 24 (AccuBioMed) using an AccuPure Tissue RNA Mini Kit (AccuBioMed). RNA purity and integrity were examined on a Bioanalyzer (Agilent), while RNA concentration was determined using a Quant-iT™ RiboGreen™ RNA assay kit and Qubit fluorometer (Invitrogen). Libraries were prepared with TruSeq Stranded mRNA HT Sample Prep Kit. Stranded, paired-end sequencing with 101 base pair read length was performed on an Illumina HiSeq2500 platform.

**Western blot**. Western blotting was performed with antibodies specific for the ERRα (#2131-1 Epitomics), YY1 (#46395, CST), PPARα (PA1-822A Thermofisher) and histone H3 (#4499, CST) proteins. In brief, snap frozen gastrocnemius muscles from 8 to 28 months-old mice (4 biological replicates for each condition) were powdered in liquid nitrogen and then homogenized in lysis buffer (20 mM Tris (pH 7.8), 137 mM NaCl, 2.7 mM KCl, 1 mM MgCl$_2$, 10 % (w/v) glycerol, 1 mM EDTA, 1 mM dithiothreitol) supplemented with protease (Roche) and phosphatase (Roche) inhibitor cocktail. After homogenization, 0.5% SDS was added in each lysate and incubated for 1 h at 4 °C with continuous rotation. The lysate was passed 10 times through a 23 gauge needle and then centrifuged at $13,000 \times g$ for 10 min. The supernatant was collected and the protein concentration was measured using the Pierce™ BCA Protein Assay kit (#23227, Thermo Scientific). 40 μg protein for each sample was separated on Mini-Protean TGX Gels, 4–20%, 10-well (# 456-1093, Biorad). The proteins were transferred on nitrocellulose membrane using semi dry apparatus (Invitrogen) for 45 min at 17 volts. The membrane was then blocked for 1 h at room temperature in 5% BSA in the TBST buffer and incubated at 4 °C overnight with continuous rocking in 1:1000 dilution of respective primary antibody in blocking solution. The next day, the membrane was washed thrice in TBST buffer for 10 min and further incubated for 1 h at room temperature with 1:10,000 dilution of horseradish peroxidase conjugated to Swine anti Rabbit secondary antibody (#P0217, Dako Denmark ApS), in blocking solution. The membrane was washed again 3 times in TBST for 10 min and subjected to visualization using ECL™ Western Blotting Detection Reagents (#RPN2016, GE Healthcare) with Fusion Fx machine (Vilber). The band intensities were quantified with the ImageJ software. Histone H3 was used as housekeeping gene and normalization control.

**Immunostaining of muscle cross sections**. Tibialis anterior muscles were mounted in optimal cutting temperature medium (O.C.T, Tissue-Tek) at resting length and snap-frozen in thawing isopentane for ~1 min before transfer to liquid nitrogen and storage at −80 °C. Muscle sections (10 μm) were cut from the mid belly at −20 °C on a cryostat (Leica, CM1950) and collected on SuperFrost Plus (VWR) adhesion slides and stored at −80 °C. Sections from both 8 and 28 months-old mice were always mounted on the same slide to ensure accurate comparisons. Sections were fixed in 4% PFA for 10 min and blocked and permeabilized in PBS containing 10% goat serum and 0.4% triton X-100 for 30 min before being incubated overnight at 4 °C in antibody solution containing 10% goat serum and antibodies against ERRα (Epitomics; 2131.1; 1:100) and Laminin α2 (4H8-2; Santa Cruz; 1:300). After washing in PBS (4 × 10 min) slides were incubated in secondary antibody solution containing Donkey anti Rat Alexa 647 (1:300; Jackson 712-605-

153), Donkey anti Rabbit Cy3 (1:300; Jackson 711-165-152) and DAPI (1 µg/ml). Muscle sections were imaged at the Biozentrum Imaging Core Facility with an Axio Scan.Z1 Slide Scanner (Zeiss) equipped with appropriate band-pass filters. Fiji macros were developed in-house to allow automated identification of muscle fibers. ERRα positive nuclei (>50% nuclei area covered by staining) were counted manually from an identical area selected from the middle of each section containing ~300 fibers using OMERO.

**Statistics and reproducibility**. All experimental measurements were expressed as mean ± standard error of the mean (SE) unless stated otherwise. Data describing mouse body and muscle measurements during aging (Fig. 1) were tested for normality and homogeneity of variance using a Shapiro-Wilk and Levene's test, respectively. One-way ANOVAs with Dunnett's post hoc tests were used to compare between 8 and 18 months data and the other four groups. Both significant differences ($p$-value < 0.05) and trends ($p$-value < 0.1) were reported where appropriate. Western blot data describing the abundance of transcription factors in mouse muscles (Fig. 7a, b) were subjected to a two-sided Mann–Whitney $U$ test between 8 and 28 months-old mice. Percentage of ERRα-positive nuclei in tibialis anterior fibers of 8 and 28 months-old mice (Fig. 7c, d) were subjected to a two-sided Student's $t$-test. All experimental data were analyzed in GraphPad Prism 8. All methods of the computational analysis with applied statistics and significance thresholds are described below in individual sections.

**RNA-Seq data sets characterizing skeletal gastrocnemius aging in mouse, rat and human**. RNA-Seq data set characterizing gastrocnemius aging in male C57BL/6JRj mice was obtained for ages 8, 18, 22, 24, 26, and 28 months with the number of replicates per age corresponding to $n = 8$, $n = 8$, $n = 8$, $n = 8$, $n = 9$, and $n = 9$, respectively. RNA-Seq data characterizing gastrocnemius aging in male Wistar rats was in part (ages of 8 ($n = 10$), 18 ($n = 10$), and 24 ($n = 9$) months) available from a previous study[12]. In addition, we also sequenced samples for 20 and 22 months with $n = 10$ replicates for each time point. The collection of rat muscle samples and sequencing followed the same experimental protocols for all time points[12]. For studying gastrocnemius aging in humans we used RNA-Seq data set obtained from the GTEx project (dbGaP accession number phs000424.v8.p2)[24]. The phenotype data table, which included the age of individuals from whom muscle samples were collected and used for sequencing, was obtained from the dbGaP annotation file of the GTEx project. To our knowledge, none of the available phenotypic measurements can be used to quantify the degree of sarcopenia. For the analysis of muscle aging in male humans we downloaded 181 gastrocnemius RNA-Seq samples from distinct individuals aged between 22 and 70 years selected by the following criteria: (i) only samples notified as "Eligible For Study" were included in the analysis; (ii) samples of only postmortem donors were considered; (iii) to exclude gender bias, we considered samples of only males; (iv) to exclude the bias induced by prolonged illnesses, only samples of individuals with the death classification "1" (violent and fast death) and "2" (fast death of natural causes) based on the 4-point Hardy scale were considered. To investigate sex-dependent differences in aging, we similarly extracted data from gastrocnemius samples of females (51 samples from distinct individuals aged between 24 and 70 years). The downloaded raw data from males were further processed with the customized workflow (see the next section). For females, we used already pre-processed data "GTEx_Analysis_2017-06-05_v8_RNASeQCv1.1.9_gene_tpm.gct.gz" available from the GTEx portal https://www.gtexportal.org/home/datasets.

For the study only biological replicates, i.e., biologically distinct samples originating from different animals/individuals, were considered.

**RNA-Seq data processing**. Paired-end RNA-Seq reads for all considered species were subjected to 3′ adapter (mate 1 5′-AGATCGGAAGAGCACACGTC-3′, mate 2 5′-AGATCGGAAGAGCGTCGTGT-3′) and poly(A)/poly(T) trimming using Cutadapt v1.9.1[81]. Reads shorter than 30 nucleotides were discarded. As reference transcriptomes, we considered sequences of protein-coding transcripts with support level 1–3 based on respective genome assemblies GRCm38 (release 92) for mouse, Rnor6.0 (release 88) for rat and GRCh38 (release 96) for human and corresponding transcript annotations from Ensembl database[82]. The kallisto v0.43.0 software was used to assign filtered reads from the rat RNA-Seq data set to the rat transcriptome, the kallisto v0.43.1 software was used to assign filtered reads from mouse and human RNA-Seq data sets to mouse and human transcriptomes, respectively[29]. The default options of kallisto were utilized for building transcriptome indices of species. For aligning stranded RNA-Seq reads for mouse, where mates 1 and 2 originated from antisense and sense strands, respectively, the option "--rf-stranded" was used. RNA-Seq reads for rat and human were obtained by unstranded protocols and did not require any parameter specification in kallisto. For all species the option "--pseudobam" was used to save kallisto pseudoalignments to a BAM file.

Mapped reads were then assigned to transcripts in a weighted manner: if a read was uniquely mapped to a transcript, then the transcript's read count was incremented by 1; if a read was mapped to $n$ different transcripts, each transcript's read count was incremented by $1/n$. Trimming 3′ adapters and poly(A)/poly(T) stretches, indexing reference transcriptomes, mapping the RNA-Seq reads to

transcripts, and counting reads assigned to individual transcripts were performed with a Snakemake framework[83].

The expression of each transcript $t_i$ was then estimated in units of transcripts per million (TPM) by dividing the read count $c_i$ corresponding to the transcript by the transcript length $l_i$ and normalizing to the library size:

$$t_i = \frac{\frac{c_i}{l_i}}{\sum_{j=1}^{\# \, of \, transcripts} \frac{c_j}{l_j}} \cdot 10^6.$$

The expression level of a gene was calculated as the sum of normalized expression levels of transcripts associated with the gene. For every gene, read counts of transcripts associated with this gene were also summed up and further used for the differential expression analysis.

**Orthology mapping**. We used the R/Bioconductor package "biomaRt" to link orthologous genes across species[84].

**Calculating the coefficient of variation of gene expression levels per age/age group**. The coefficient of variation (CV) is a measure of dispersion of a distribution, often used to assess the gene expression heterogeneity among replicate samples. We have computed the CVs for our data sets to assess the heterogeneity in gene expression as a function of age. For each organism, age/age group, and gene, we calculated the mean value of the expression (in TPM units) and standard deviation of the expression across replicates. We then computed the coefficient of variation (CV) as the ratio of the standard deviation to the mean. Then, for each organism and each age/age group we plotted the distribution of CVs across all genes in the form of violin plots (Fig. 2d–f). We took the median value of CVs for the youngest age/age group as a baseline (thin blue line drawn across all ages/age groups). Median values of CVs for other ages/age groups (thick blue lines within violin plots) are mostly located over the baseline, especially for older ages/age groups, indicating higher heterogeneity across replicates for these ages/age groups in comparison to the youngest one.

**Differential expression analysis**. Differential expression analysis was performed with EdgeR available through the R/Bioconductor package[31] for each species separately. For rodents, a gene was included in the analysis only if it had at least 1 count per million (CPM) in the number of samples corresponding to the minimum number of replicates of the same age across ages. For humans, a gene was included in the analysis only if it had at least 1 count per million (CPM) in the number of samples corresponding to the minimum number of replicates across age groups (20–29, 30–39, 40–49, 50–59, 60–69, and 70–79 years). Gene expression was considered statistically different between two conditions, if the false discovery rate (FDR) was less than 0.01[85].

**Aligning gene expression with principal components**. Calculating principal components (PCs) for RNA-Seq data sets and aligning gene expression with a PC was performed individually for each organism. Here, we described the general procedure applied for each species, respectively.

The gene expression matrix with samples as columns and log2-transformed gene expression in TPM units as rows was mean centered to make the data comparable both across samples and genes. The centered gene expression matrix was further subjected to the principal component analysis (PCA). First two principal components, PC1 and PC2, were defined for each species, respectively (Fig. 2a–c).

Then for each organism we quantified how much individual genes contributed to the corresponding PC1[35,36]. To explain the concept of the analysis, we created a toy data set consisting of three "samples" and 26 "genes" measured for each sample. Gene expression in these samples was randomly sampled from the multivariate normal distribution and stored in the form of the mean centered gene expression matrix described above. Supplementary Fig. 6a visualized genes from the toy data set as dots located in the sample space, where coordinates of genes in this space were defined by their expression in corresponding samples. Genes can also be represented as vectors in the same space, where rows of the mean centered gene expression matrix correspond to the coordinates of these vectors (a blue vector associated with a gene). Further, the direction of the highest variance (PC1) and the direction of the second highest variance orthogonal to PC1 (PC2) were defined in the sample space (black vectors). For determining the contribution of individual genes to the PC1, we calculated the magnitude of the projection of gene vectors on the PC1 (dashed red line). Genes with extreme-positive and extreme-negative projections contribute most to the PC1.

Projections of gene vectors on PCs are also known as "weights"/"loads"/"loadings". PCs create a new basis for genes located in sample space. Thus, the expression of a gene $g_i$ in samples $s_1, s_2, \dots, s_n$ can be represented as the linear combination of principal components PC1, PC2, ..., PCn:

$$g_i = p_{i1}PC1 + p_{i2}PC2 + \dots + p_{in}PCn,$$

where $p_{i1}$ is the coordinate of $g_i$ on PC1 defined by the projection of $g_i$ on PC1, $p_{i2}$ is the coordinate of $g_i$ on PC2 defined by the projection of $g_i$ on PC2 and so on. Coordinates $p_{i1}, p_{i2}, \dots, p_{in}$ of the gene $g_i$ in the space of PCs are also called

"weights"/"loads"/"loadings". For the analysis, we used projections of expressed genes on PC1, i.e., $\{p_{i1}\}_{i=\overline{1,m}}$, where $m$ corresponds to the number of genes.

Analogously to the toy example, for each species we defined the space of all samples from the RNA-seq data set. Coordinates of genes in this space corresponded to their expression in each sample based on log2-transformed and mean centered gene expression matrix. Then PC1 in this space was localized, projections of gene vectors on PC1 were calculated and used for further analysis. Note that projections of gene vectors on PC1 are normally distributed for all considered species (Supplementary Fig. 6b). This stems from the fact that log2-transformed normalized gene expression levels in TPM units are close to normally distributed[86]. Mean centering and projecting data on a vector (in our case it is PC1) are linear transformations, which preserve data distribution.

For the comparison across species, we standardized the projection values by calculating corresponding z-scores (Supplementary Fig. 6c). Expression of genes with extreme, positive projection z-scores increased during the aging-related loss of muscle mass and functionality, whereas that of genes with extreme, negative projection z-scores decreased.

Note that only expressed genes were used for calculating PCs and defining their contribution to PCs. For rodents, a gene was considered as expressed if it had at least 1 transcript per million (TPM) in the number of samples corresponding to the minimum number of replicates of the same age across ages. For humans, a gene was considered as expressed if it had at least 1 TPM in the number of samples corresponding to the minimum number of replicates across age groups (20–29, 30–39, 40–49, 50–59, 60–69, and 70–79 years).

**Gene set enrichment analysis**. We applied gene set enrichment analysis (GSEA) to define pathways involved in gastrocnemius aging of mouse, rat, and human[37]. Pathways were composed of genes based on the KEGG database (http://www.kegg.jp)[38]. Figure 4d demonstrates the concept of the analysis using as an example the gastrocnemius time course RNA-seq data for mouse and two pathways "Oxidative phosphorylation" and "Jak-STAT signaling pathway". The analysis for other pathways and other organisms was performed in a similar way. For each organism expressed genes (see the above section) were ranked by their projection z-scores on PC1. For each pathway and each organism, the enrichment score (ES) was calculated reflecting the degree to which pathway genes are overrepresented at the top or bottom of the ranked gene list. A positive ES indicates the pathway enrichment by genes with positive projection z-scores like "Jak-STAT signaling pathway", whereas a negative ES indicates the pathway enrichment by genes with negative projection z-scores like "Oxidative phosphorylation". In addition to the ES score, the normalized enrichment score (NES) accounting for differences in pathway sizes and false discovery rate (FDR) accounting for the significance of the NES score were calculated. Details how NES scores and FDRs are calculated can be found here[37]. A pathway was reported if the corresponding false discovery rate (FDR) was less than 0.01 for at least one species (Fig. 4e). For visualizing GSEA results we considered both FDR and normalized enrichment score (NES) indicating if the enrichment was based on upregulated (NES > 0) or downregulated (NES < 0) genes. For the hierarchical clustering of reported pathways, we used Euclidean distance for significance measures 1-FDR for pathways with NES > 0 and FDR-1 for pathways with NES < 0. A pathway was considered to be significantly enriched for all species if the FDR was less than 0.1 for all species (Fig. 5a).

For pathways significantly enriched for all species we defined genes constituting the leading edge, i.e., genes contributing most to the ES. For a positive ES (like "Jak-STAT signaling pathway"), the leading edge includes genes that appear in the ranked gene list prior to the peak score (Fig. 4d, red bars in the dark red box). For a negative ES (like "Oxidative phosphorylation"), the leading edge comprises genes that appear in the ranked list subsequent to the peak score (Fig. 4d, blue bars in the dark blue box).

**Estimating transcription factor and miRNA activities**. We used ISMARA tool[39] to estimate the activity of transcription factors (TFs) and miRNAs during skeletal muscle aging across species, and to identify their most likely targets. We were able to analyze the activity of 584, 502, and 603 motifs corresponding to TFs/miRNAs in mouse, rat and human, respectively, one motif frequently corresponding to multiple TFs/miRNAs. In brief, ISMARA models the expression levels of all mRNAs in a sample in terms of the activity of all TFs and miRNAs, and the predicted response of these mRNAs to the regulators.

To identify TFs/miRNAs regulating gene expression during muscle aging, we checked how many of the leading-edge genes from pathways commonly regulated across species (Fig. 5a) were among the ISMARA-predicted targets of individual TFs/miRNAs. We carried out the analysis separately for pathways where the expression of leading-edge genes decreased during muscle aging (e.g., "Oxidative phosphorylation") and pathways where the expression of leading-edge genes increased (e.g., "Jak-STAT signaling pathway"). For each organism and each of the two groups of pathways, the top 10 TFs/miRNAs predicted to have most leading-edge genes among their top 300 targets were identified. We then matched the symbols corresponding to the regulators to find those that are shared between species (Fig. 6a, b). For visualizing the expression of the top 300 targets of conserved TFs (Fig. 6c–e and Supplementary Figs. 8 and 9), we normalized the data in the same way as we did when we calculated the slopes of gene expression changes (Supplementary Fig. 7c). Two-sided Student's t-test was applied to TF activities to test the difference between the youngest age/age group and all other

ages/age groups. The difference with a p-value less than 0.05 was considered as significant.

**Gene ontology analysis**. To annotate genes differentially expressed between groups "G1" and "G2" of 28-months-old mice (Fig. 3b, dashed ellipses), we performed the gene ontology (GO) analysis using Database for Annotation, Visualization and Integrated Discovery (DAVID)[87] through the R/Bioconductor package called "RDAVIDWebService"[88] (Supplementary Fig. 4). "GOTERM_BP_DIRECT", "GOTERM_MF_DIRECT", and "GOTERM_CC_DIRECT" categories were used for gene annotation. Similarly, we characterized processes regulated by conserved TFs, where we subjected their top 300 targets (Fig. 6c–e and Supplementary Figs. 8, 9) to the GO analysis (Fig. 6i–k and Supplementary Figs. 8, 9). Background genes for calculating enrichment statistics were defined for each species separately and consisted of all genes expressed in muscle samples (see the section "Aligning gene expression with principal components"). GO terms with a p-value less than 0.01 were considered significantly enriched.

**Shiny application**. To make our rodent high-throughput data sets and data analysis tools available for the research community, we included them in the previously developed interactive web application "SarcoAtlas" based on the R package "Shiny" (version 0.14.2, https://cran.r-project.org/web/packages/shiny/index.html). For each data set the application supports gene expression plotting, differential expression analysis, principal component analysis, and aligning gene expression with principal components. Moreover, the application can submit genes resulting from the analysis to STRING[89] to further investigate protein-protein interactions and perform GO analysis. The application can be accessed through the following link: https://sarcoatlas.scicore.unibas.ch/[25].

**Reporting summary**. Further information on research design is available in the Nature Research Reporting Summary linked to this article.

## Data availability

RNA-Seq data set describing gastrocnemius aging in male C57BL/6JRj mice was deposited to Gene Expression Omnibus (GEO, https://www.ncbi.nlm.nih.gov/geo/)[90] under the accession number GSE145480. RNA-Seq data set characterizing gastrocnemius aging in male Wistar rats was submitted to GEO under accession numbers GSE78702 and GSE146976 (series GSE146977). For studying gastrocnemius aging in humans we used RNA-Seq data set obtained from the GTEx project (dbGaP accession number phs000424.v8.p2)[24]. The phenotype data table including the age of individuals from whom muscle samples were collected and used for sequencing was obtained from the dbGaP annotation file of the GTEx project and has restricted access. Experimental measurements in rodents underlying the graphs and charts are available as Supplementary Data 1.

## Code availability

The source code to replicate the analysis presented in this study is available from Zenodo at https://doi.org/10.5281/zenodo.375417[91].

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

## Acknowledgements

We gratefully acknowledge Dr. Christoph Handschin for help with funding acquisition, Dr. Mikhail Pachkov for help with data processing, Dr. Erik van Nimwegen and Dr. Jeremie Breda for fruitful discussions about data analysis, Beatrice Dimitriades and Lea Mues for their help in accomplishing experiments and Christina Herrmann for help with the figure design. We also acknowledge the support of the University of Basel's Quantitative Genomics Facility, in particular Phillippe Demougin for assistance with RNA-Seq sample preparation, and the Biozentrum Imaging Core Facility. Calculations were performed at sciCORE (http://scicore.unibas.ch/) scientific computing center at University of Basel. This work was supported by the Cantons of Basel-Stadt Basel-Landschaft, a Sinergia grant (CRSII3_160760) from the Swiss National Science Foundation awarded to M.A.R., M.Z., and Christoph Handschin and a Novartis University of Basel Excellence Scholarship for Life Sciences (3BZ8003) awarded to A.B.

## Author contributions

M.Z., A.B., M.A.R., D.J.H., L.A.T., and N.M. conceptualized the study. M.Z. and M.A.R. supervised the study. D.J.H. and M.A.R. performed the mouse study design. A.B. and M.Z. designed the analysis. D.J.H. collected and analyzed mouse samples, performed microscopy imaging and image analysis. N.M. and D.J.H. did sample preparation for mouse mRNA sequencing data. A.B. performed all computational analyses. N.M. measured protein abundance by western blotting. E.M. and J.N.F. collected and analyzed rat samples. A.B. and D.J.H. prepared the figures. A.B., M.Z., and D.J.H. wrote the original draft. All authors read and improved the final manuscript.

## Competing interests

Authors E.M. and J.N.F. declare the following competing interests: they are full-time employees of Nestlé SA. All other authors declare no competing interests.
