## [Peer Review File · Communications Biology]

Reviewers' comments:

Reviewer #1 (Remarks to the Author):

Börsch and colleagues have put together a well-written and highly-relevant contribution to the field, covering a time series of phenotypic measurements and RNA-Seq data from gastrocnemius of mouse and comparing these to rat, and human publicly available RNA-Seq data. The authors tease out commonalities in pathways that are altered due to ageing across all three species (mitochondrial function), and transcription factors that have translational and biomarker potential. In addition, the authors describe muscle mass as a stronger predictor of ageing and muscle health, than age itself. It was interesting to see that mice showed the same hind-limb predominant lean muscle loss as previously described in literature for humans.

Overall, this study has been very well constructed, the experiments were well conducted, and the manuscript has been a pleasure to read. Given the predominant use of mouse and rat models in ageing studies and studies involving muscle wasting, the findings of this study are of significant clinical relevance.

However, these are some issues I think the authors may want to address:

1. Abstract: Although the authors end on the note that phenotypic measurements should be considered in analyzing ageing-related molecular data, the finding that phenotypic measurements rather than age itself are a stronger predictor of ageing in muscle, are a highly relevant finding to the field and should be emphasized more.
Furthermore, it is interesting and reassuring to see that mice follow the same pattern of weight loss across the lifespan as humans, with a predominant lean tissue decline in hind-limbs (lower-limbs in humans), and less fat loss around internal organs and general abdominal area. This commonality with humans deserves to also be pointed out in the abstract given its clinical relevance to human.
2. Can you speculate why the TA is a notable exception to age-related muscle loss?
Previous studies show TA is affected by sarcopenia in rat, as opposed to the human. (For example see doi: 10.18632/aging.100926, and citations)
3. Is data for skeletal muscle index available within this data base? If so, correlation of PC1 with SMI would be of great interest.
4. If I understood correctly, the authors tested muscle function in EDL and SOL. I recommend that the authors specify why EDL and SOL were picked specifically and not gastrocnemius for functional tests, given RNA-Seq was conducted on the latter. Mixed fibre composition and the focus thereon on gastrocnemius is discussed in the Discussion section: SOL consists almost exclusively from type I fibres in all three species, and EDL is predominantly type IIb in all three species, whereas gastrocnemius is ~50% type I, with type II fibres being most susceptible to ageing. I can see the reasoning for showing EDL and not SOL functional decline, I suggest this should also be explained.
5. These results recapitulate the molecular changes observed in "male" human sarcopenia, this is something I recommend the authors stress in the manuscript and highlight particularly at the end of the discussion. Furthermore, I would expect authors to highlight sex differences, especially since differential sex specific rate of absolute muscle loss has been suggested and described in previous literature. (For example, see doi: 10.1007/s11357-015-9860-3 and literature within)
6. I understand further experiments on female mice/rats would be too much to ask. However, if authors would want to generalize the study on overall human sarcopenia, a follow up analysis on female RNA-Seq data and overlap with male data (after commonalities have been found with other species), would hugely benefit the overall conclusions.

Minor comments:

(Page 1, line 14, Abstract): Please add "age related" so it reads "Age related loss of skeletal muscle..." for appropriate definition of sarcopenia.

(Page 2, line 81, Results): How is healthy weight range defined in mice? Please consider adding literature or a definition.

(Page 3, line 89-91, Results): You have shown a great overview of % fat-loss, % reduction in tetanic force, % decrease in limb grip strength throughout this paragraph. Please consider including overall % loss of lean mass across hind-limb muscles in this paragraph, to help readers with overall comparisons.

(Page 3, line 91-92, Results): I might misunderstand this figure. As I understand it, it shows loss of muscle mass relative to body weight at 28 months, normalized to 8-18 months. This highlights the predominant focus of muscle wasting on hind-limbs, this can be emphasized here and the overall conclusion (Page 3, line 106-109, Results). Given the decrease in lower skeletal muscle mass in humans associated with ageing this should be of great interest for the readership (For example see <https://doi.org/10.1152/jappl.2000.89.1.81> - Skeletal muscle mass and distribution in 468 men and women aged 18-88 yr).

Have you assessed whether loss of muscle mass outstrips loss of body mass overall? E.g. fat mass, liver, heart and epi. fat in the same graph and % loss of mass relative to body weight – I'd presume this would even further strengthen the point that lean mass decreases most across hind-limbs in mouse, drawing a parallel to the human.

(Page 4, line 128-129, Results): For more clarity I recommend the authors make the following change:

"and young animals we observed changes in the expression of numerous genes (Supplementary Fig. 1), however, the vast majority of changes were smaller than 2-fold (Fig. 2g-h)."

Figure 1 legend

(Page 25, line 882) a > (a) (as for b, c, d, e & f in figure legend)

Figure 1 b: Lean mass > centered

Figure 1e: For more clarity I recommend the authors use the following labelling of Figure 1e.

"Loss of muscle mass relative to body weight at 28 mos (normalised to 8-18 mos, %)"

With the presumption that m was meant to abbreviate months, use the abbreviation mos.

Figure 1 g-j: I recommend the authors add EDL to vertical labelling of figures, so it's clear within the figure that these recordings have been undertaken on EDL alone.

Figure 2 legend

(Page 25, line 896) a-c > (a-c) (as for d-f, g-i in figure legend)

Figure 3 legend

(Page 25, line 908) a > (a) (as for b-c in figure legend)

(Page 25, line 911) p-value0.05 change to p-value \geq 0.05

Figure 4 legend

(Page 25, line 917) a-c > (a-c) (as for d in figure legend)

(Page 25, line 919) I think the authors should show that 'mmu', 'mno' and 'hsa' abbreviations stand for = mus musculus, rattus norvegicus and homo sapiens.

Figure 5 legend

(Page 26, line 928) a > (a) (as for b in figure legend)

Figure 5 a legend

Please consider providing information to help readers interpret Figure 5a, currently this reads more like just the methodology used.

Figure 6 legend

(Page 26, line 935) a > (a) (as for b, c-d in figure legend)

Figure 6, Supplementary Figure 8 and 9 – second column:

I might have missed this, but between which groups has the students t-test been performed and why? With multiple age groups I would expect to see comparison of the youngest with each of the other groups as undertaken previously.

Supplementary Figures:

A light grey border appears around the rat and mouse schematics when using the zoom in/out function in supplementary figures. See screenshot attached, you might want to reformat the figures.

Supplementary Figure 6:

"... (blue vector) on PC1 (black vector)" you mean "... (blue vector) on PC1 (blue vector)"? (PC1 is blue)

Signed: Tom Gillingwater / Ines Boehm

Reviewer #2 (Remarks to the Author):

In the manuscript by Börsch, Ham et al., the authors employ a comprehensive phenotypic and bioinformatic approach to identify molecular signatures of age-associated loss of skeletal muscle mass and function i.e. sarcopenia. By utilizing three different model systems (human, rats and mice), they sought to identify common transcriptional signatures that are associated with muscle aging. In addition, the authors make use of the temporal information in the data sets in order to attempt to uncover novel pathways that could be driving the loss in muscle function with age. Their computational pipeline leads to a number of molecular pathways that are conserved across species during aging, most notably the loss of mitochondrial function and increased inflammation. Lastly, their datasets provide insights into which rodent model should be used depending on what biological process/pathway you are studying, particularly if wanting to translate the findings into a human population. Overall, the manuscript is well-written and the computational approach is quite sophisticated, with their results and 'SarcoAtlas' undoubtedly being a valuable resource for researchers studying muscle biology during aging. My specific comments are listed below:

Major Comment

- The authors did a very nice job of utilizing the temporal resolution in their data sets to identify potential drivers of muscle aging; however, the results from this computational approach would have benefited greatly from a final validation step showing differences in the abundance/activity of the TFs

that are proposed to contribute to the differences in the transcriptional signature. This would add strength and confidence that the computational approach was able to uncover novel targets involved in skeletal muscle aging. This could be done just in the rodent models, since tissue samples are easier to obtain.

Minor Comments

- I'm curious as to why the significant drop in muscle mass is not associated with a significant drop in total lean mass, particularly since the authors point out that skeletal muscle can contribute to up to 50% of body weight. Are other organs making up the difference?
- The authors mention that they used the gastrocnemius from the rodents because there was corresponding data in the human population. Is there any phenotypic data for the human population? Could this possibly explain why there is such a small number of differentially expressed genes in the human dataset? It's possible that the muscle health of these subjects was relatively normal, and that was why the transcriptional profile did not differ that much from the young counterparts.
- Similarly, the authors should comment on how representative the respective age groups of the model systems are to one another. This fact was touched on a bit, but it should be mentioned that the human age group is still relatively "young" compared to the mice, who are extremely old. Without phenotypic data, it's hard to know whether the human population is sarcopenic, thus complicating the interpretation.

Reviewer #3 (Remarks to the Author):

Brief summary of the manuscript

Börsch et al. analyzed and compared skeletal muscle phenotypic and molecular parameters and their age-projections between rodent animal models and humans with the goal to evaluate the usefulness of animal models to recapitulate human skeletal muscle aging. The authors based their conclusions on skeletal muscle and whole body phenotypic data (collected for mice in this study; body mass, muscle mass and function), and molecular data (all three species, RNAseq from mouse (this study), rat (this and previously published study), and human (using a public database)) over a spread of ages. Gene and pathway correlations between species were analyzed with statistical methods and models. The main finding of this study is that rodents indeed seem to recapitulate human muscle aging well, but the authors qualified that one or the other rodent species better represents specific aspects of human muscle aging (such as inflammation or metabolism). Further, 1) human muscle aging seems to be recapitulated by rodent models on the pathway rather than the individual gene level, and 2) the timing of muscle aging, reflected in changes in pathway regulation, differs between rodents and humans.

Overall, this study represents an interesting body of analytical and computational work that transforms individual gene expression analysis across species into a correlation analysis of molecular pathways. The authors created and utilized existing large data sets and comprehensively assessed skeletal muscle aging in this cross sectional study. Interestingly, the authors found no key driver of skeletal muscle aging on the individual gene level, no obvious strong correlation between "skeletal muscle health" and chronological age, and only a handful of differentially expressed individual genes overlapping between the species, diminishing enthusiasm in limited gene expression preclinical studies for the evaluation of human aging. Molecular data seem to correlate better than chronological age with phenotypic data, with pathways correlating better between the species and with aging than individual gene expression data. The authors further present an array of differentially expressed molecular pathways shared between at least two species as future target pathways for rodent studies recapitulating human skeletal muscle aging. What is furthermore interesting is the report of age-

related changes in transcription factor activities.

The findings gained from these analyses could have significant implications for the design and interpretation of future studies investigating muscle aging in animal models as well as in humans. In addition, the authors' data and main conclusions have built a basis for future investigations into the regulation of muscle aging and potential interventional targets at the level of transcriptional regulators. It is therefore of interest to the wider field of aging biology on both the preclinical and the clinical level.

The authors define in the methods section the way the data are calculated; the datasets are accessible; and for a researcher well versed in the respective computational methods, the data should be reproducible. Nevertheless, it is evident that the authors speak to a subset of aging researchers with more extensive knowledge of statistical analyses and modeling. The authors explained (at least for the most part) how they derived the PCA data and subsequent computations, using "toy data sets". However, although PCA of large data sets such as RNAseq data is not uncommon, the results and additional data derivations such as coefficient of variance analysis (Fig 2d-f) presented here are not always sufficiently explained for the wider target audience that could and would want to take advantage of the authors' findings and conclusions. It is not completely obvious why specific analyses were chosen, how the resulting data were derived, and what can be learned from these derivations. The reader would benefit from more explanation (without creating a statistics textbook). This shortcoming prevents this reviewer and possibly a wider audience from fully evaluating the validity of the results, appreciating the conclusions drawn from them, and implementing the presented insights into future study design and analyses.

Specifically:

Figure 2a-c: The PCA of the molecular RNAseq data revealed that PC1 in the three species explains 37 to 18% of the variance between samples within each species and PC2 8 to 14% (Fig 2a-c). However, it is not elaborated on which major genes comprise PC1 and PC2, and what their respective weights (or loads) are. Is there any potential information to be gained from disclosing the genes that provide most weight to the PC1 (and 2) and for the nature of the genes' loading scores? If so, please add, if not, please explain why not.

Figure 2d-f: The next piece of information given is plotted in Figure 2d-f, which aims to show the scatter of gene expressions along PC1 and 2, and potential age-clustering (or the lack thereof). The authors claim that the variability increases with age (Fig. 2d-f); however, without more explanation of these plots (Fig. 2d-f) and their construction, and the description in the results section (line 127-127) it remains rather obscure (what am I looking at, where is the higher variation in older groups compared to younger, what do the black lines/outlines mean?). Understanding these data will help appreciating the conclusion, that chronological age does not necessarily correlate with gene expression changes.

Figure 3: The authors went on to show that the PC1 correlate with phenotypic data better than with age. This is not entirely obvious from the Figure 3a, since the correlation coefficients between PC1 and age versus PC1 and muscle mass do not seem to differ much (both $\sim 0.7-0.75$). However, this claim seems more apparent in Fig. 3b. Please clarify this discrepancy between Fig 3a and 3b? Are the graphs in the Supplementary Figure 2 and 3 (age vs PC1) are the same data shown in Fig 3b?

Figure 4: Figure 4 builds on projection values and their standardization by z-scores. For clarification (from which the reader would likely benefit as well): does the magnitude of the gene's projection vector onto PC1 correspond to this variable's load? Further, please add why normal distribution of the

projection of gene vectors onto PC1 was expected (line 177)?

Figure 6 (particularly 6 c-e and respective supplementary figures) is not well explained. Is any of the TF activities significantly different between ages or age-groups (only human age group 20-29 is indicated to be significantly different from 30-39)? The third column in Figure 6 c-e needs to be clarified: enriched GO terms, meaning TF activity associated with those pathways decrease (as indicated in the other two columns)?.

Line 362: was PCG1a also found downregulated in the three data sets?

Line 312 and following: reference to the NIA Rapamycin study could be useful (PMID: 19587680).

The comments of the reviewers are highlighted in blue color and numbered in italics according to the following scheme: the number before the dot corresponds to the reviewer's ID, i.e. 1, 2 and 3, and the number after the dot indicates the index of the comment made by the respective reviewer. Our response to each of the comments follows in black font. The line numbers are shown for the revised manuscript. All changes made in the revised manuscript are highlighted with yellow color.

Reviewer #1 (Remarks to the Author):

Börsch and colleagues have put together a well-written and highly-relevant contribution to the field, covering a time series of phenotypic measurements and RNA-Seq data from gastrocnemius of mouse and comparing these to rat, and human publicly available RNA-Seq data. The authors tease out commonalities in pathways that are altered due to ageing across all three species (mitochondrial function), and transcription factors that have translational and biomarker potential. In addition, the authors describe muscle mass as a stronger predictor of ageing and muscle health, than age itself. It was interesting to see that mice showed the same hind-limb predominant lean muscle loss as previously described in literature for humans.

Overall, this study has been very well constructed, the experiments were well conducted, and the manuscript has been a pleasure to read. Given the predominant use of mouse and rat models in ageing studies and studies involving muscle wasting, the findings of this study are of significant clinical relevance.

We thank the reviewers for these comments.

However, these are some issues I think the authors may want to address:

1.1 1. Abstract: Although the authors end on the note that phenotypic measurements should be considered in analyzing ageing-related molecular data, the finding that phenotypic measurements rather than age itself are a stronger predictor of ageing in muscle, are a highly relevant finding to the field and should be emphasized more.

We thank the reviewers for this suggestion. We agree that this is an important point and have stated it more explicitly in the abstract (while remaining within the allowed word count). The final sentence of the abstract now reads (see l. 25-27, page 1):

“Our study demonstrates that phenotypic measurements such as the muscle mass, are better indicators of muscle health than the chronological age and should be considered when analyzing aging-related molecular data.”

Furthermore, it is interesting and reassuring to see that mice follow the same pattern of weight loss across the lifespan as humans, with a predominant lean tissue decline in hind-limbs (lower-limbs in humans), and less fat loss around internal organs and general abdominal area. This commonality with humans deserves to also be pointed out in the abstract given its clinical relevance to human.

We thank the reviewer for this suggestion as well. The limit on abstract word count is really low and we have therefore included this important point in the results section (see l. 126-128, page 4):

“Importantly, the temporal pattern and magnitude of reduction in skeletal muscle mass and function reach a level consistent with the clinical definition of sarcopenia²⁸ at 28 months of age.”

1.2 2. Can you speculate why the TA is a notable exception to age-related muscle loss?

Previous studies show TA is affected by sarcopenia in rat, as opposed to the human. (For example see doi: 10.18632/aging.100926, and citations)

We thank the reviewers for a very interesting point. First, we note that the *tibialis anterior* does experience a significant age-related loss in mice as well (updated Fig. 1d, see below). However, in contrast to the other measured muscles, this loss can be entirely explained by the age-related loss of body mass (as can now be seen more clearly in the updated Fig. 1e, see below). While investigating inter-muscle differences in sarcopenia was not a focus of the current study, as comparative data on multiple muscles are not available, we recently explored this question in more detail in mouse (Ham and Börsch, Nature Communications, 2020, <https://doi.org/10.1038/s41467-020-18140-1>) by comparing RNA-Seq profiles from *tibialis anterior*, *gastrocnemius*, *soleus* and *triceps brachii* muscles in young and old mice as well as old mice that underwent a long-term treatment with rapamycin. In this study we found that *tibialis anterior* had a lower age-related muscle loss than *gastrocnemius*, though the genes in the aging-relating signature were similar. The rapamycin treatment further revealed differences between these two muscles (Fig. 2a in the mentioned study), as *tibialis anterior* had a heightened ‘denervation response’ gene signature (Fig. 7e in the mentioned study) and was better protected from denervation-induced wasting than the *gastrocnemius* muscle (Fig. 7f in the mentioned study).

We have highlighted this inter-muscle difference more clearly and refer interested readers to our recently published work (see l. 100-102, page 3):

“Muscle loss outstripped body mass loss for all measured muscles, with the notable exception of the tibialis anterior, which was comparatively more resistant to age-related muscle loss, as previously observed in both humans¹² and mice²⁷.”

Fig. 1. Muscle mass and function progressively decline in male C57BL/6JRj mice during aging. a Body mass for 8, 14, 18, 22, 24, 26 and 28 months-old mouse groups. EchoMRI measurements of

b whole-body lean and **c** fat mass. **d** Absolute muscle mass for *quadriceps* (QUAD), *gastrocnemius* (GAS), *tibialis anterior* (TA), *plantaris* (PLA), *extensor digitorum longus* (EDL) and *soleus* (SOL) averaged across both limbs. **e** Body, lean and fat mass as well as muscle tissue and organ mass in 28 months-old mice normalized to the mean of 8, 14 and 18 months-old groups. The mean percentage loss in mass relative to body mass is reported above each data set. The color scheme designates the direction of changes and significance: grey is not different (p-value>0.10), red is increased (p-value<0.05), blue is decreased (p-value<0.05), light red is a trend for increased (0.05<p-value<0.10). **f** Recordings of all-limb grip strength. Isolated EDL muscle function parameters, including **g** force-frequency curve (left) and fatigue response to multiple stimulations (right); and twitch time-to-peak tension in **h** and half relaxation time in **i**. Group numbers are: 913 in **a-f**; 7-13 in **g** (left); 5-12 in **g** (right) and 8-13 in **i-j**. For statistical comparisons 8, 14 and 18 months-old groups were pooled and compared with each of the other four groups. *, ** and *** denote a significant difference between groups of p-value<0.05, p-value<0.01 and p-value<0.001, respectively. Trends (0.05<p-value<0.10) are denoted by # or the p-value specified. Colored asterisks refer to the group of comparison.

1.3 3. Is data for skeletal muscle index available within this data base? If so, correlation of PCI with SMI would be of great interest.

We agree with reviewers that the availability of phenotypic measurements reflecting muscle health of humans (e.g. skeletal muscle index) would strengthen the analysis. Even though GTEx provides quite extensive phenotypic data for the human population (see https://www.ncbi.nlm.nih.gov/projects/gap/cgi-bin/dataset.cgi?study_id=phs000424.v8.p2&phv=169091&phd=3910&pha=&pht=2742&phvf=&phdf=&phaf=&phtf=&dssp=1&consent=&temp=1), these measurements are generally global (e.g. blood parameters), rather than related to organs/tissues. In particular, skeletal muscle index is also not available and therefore, we cannot determine whether/which human individuals were sarcopenic. We clarified this in l. 602-603, page 15:

“To our knowledge, none of the available phenotypic measurements can be used to quantify the degree of sarcopenia.”

1.4 4. If I understood correctly, the authors tested muscle function in EDL and SOL. I recommend that the authors specify why EDL and SOL were picked specifically and not gastrocnemius for functional tests, given RNA-Seq was conducted on the latter. Mixed fibre composition and the focus thereon on gastrocnemius is discussed in the Discussion section: SOL consists almost exclusively from type I fibres in all three species, and EDL is predominantly type IIb in all three species, whereas gastrocnemius is ~50% type I, with type II fibres being most susceptible to ageing. I can see the reasoning for showing EDL and not SOL functional decline, I suggest this should also be explained.

The *extensor digitorum longus* (EDL) muscle was chosen for in vitro muscle function testing because of its physical properties. Specifically, the EDL has distinct distal and proximal tendons allowing the muscle to be isolated and secured to the force motor arm and it is thin enough to allow effective diffusion of nutrients (e.g. oxygen) within an organ bath. It is possible to perform in situ

muscle function on the *gastrocnemius*, but for this experiment the mouse must undergo a 30-45 min stimulation procedure before muscle collection which could potentially influence signaling pathways. Since assessing the activity of pathways was the primary aim of the study, we decided to use isolated EDL function as representative of fast-twitch muscle function. Importantly, we also show that reductions in all-limb grip strength (Fig. 1f) mirrors the age-related loss of muscle mass. In accordance with the reviewers' request, we have updated the text to include the rationale for the use of the EDL (see l. 112-115, page 3):

“Next, we isolated the extensor digitorum longus (EDL) muscle to test muscle function directly (Fig. 1g-i). The EDL is a fast twitch, hindlimb muscle that can be isolated tendon-to-tendon and is thin enough for effective nutrient diffusion in an organ bath.”

1.5 5. These results recapitulate the molecular changes observed in “male” human sarcopenia, this is something I recommend the authors stress in the manuscript and highlight particularly at the end of the discussion. Furthermore, I would expect authors to highlight sex differences, especially since differential sex specific rate of absolute muscle loss has been suggested and described in previous literature. (For example, see doi: 10.1007/s11357-015-9860-3 and literature within)

Indeed, we chose to analyze data from males, for the following reasons. First, sequencing data for male rats were available prior to our study. Second, at the time of our study aged female mice were not commercially available in Europe. We included a paragraph at the end of the Discussion section pointing out that our study is dedicated to investigating ‘male’ sarcopenia and that according to the literature there might be sex-specific differences in the emergence and development of sarcopenia not covered by our analysis (see l. 476-481, page 12):

“An outstanding question concerns sex-specific differences in the emergence and development of sarcopenia. Previous studies suggested that the rate of absolute muscle loss during aging is sex-specific⁸⁷ and reported sex-specific risk factors for sarcopenia such as malnutrition in females and higher serum myostatin in males⁸⁸. However, the molecular-level differences between genders during muscle aging still remained unclear. Our study focused on characterizing sarcopenia in males, due to the availability of data in rats and of aged mice.”

1.6 6. I understand further experiments on female mice/rats would be too much to ask. However, if authors would want to generalize the study on overall human sarcopenia, a follow up analysis on female RNA-Seq data and overlap with male data (after commonalities have been found with other species), would hugely benefit the overall conclusions.

As discussed above, for reasons of material and data availability we have focused our study on male sarcopenia. However, to address this important point raised by the reviewer, we included in our revised manuscript an analysis of gene expression changes occurring during muscle aging in human females and males (Supplementary Fig. 12, see below). The analysis demonstrated that changes in the gene expression in the *gastrocnemius* muscle of females during aging correlated strongly with changes occurring in the same muscle of males. Moreover, GSEA demonstrated similar enrichment for KEGG pathways whose activity changes during muscle aging in males and females. Thus, our study suggests a high degree of similarity in molecular changes occurring in *gastrocnemius* muscle of female and male humans during aging. We mention that further studies

integrating estimates of muscle health and molecular measurements in skeletal muscles across different ages are required to conclusively identify sex-specific differences. The results of the analysis are presented in Supplementary Fig. 12 and were summarized in l. 482-490, page 12:

“Nevertheless, we attempted to address this issue in humans, by analyzing RNA-Seq samples from the GTEx database for females and males separately (Supplementary Fig. 12). The gene-level analysis showed a highly similar response to aging in both females and males, with age-related gene expression changes in the gastrocnemius strongly correlated ($r=0.91$) between male and females. Likewise, the molecular pathways changing their activity during muscle aging followed the same pattern for both female and male humans. Thus, although future work is needed to more conclusively answer the question of sex-specific patterns of muscle aging, our study indicates a high degree of similarity in the molecular changes occurring in female and male human gastrocnemius muscle at high age.”

The selection of *gastrocnemius* samples in female humans followed the same criteria as for male humans. For the analysis we used pre-processed data available from the GTEx portal (see Methods section, l. 610-618, page 15):

“The same criteria were applied for the selection of gastrocnemius samples of female humans (51 samples from distinct individuals aged between 24 and 70 years). The downloaded raw data of male humans were further processed with the customized workflow (see the next section). To investigate sex-dependent differences in aging, we similarly extracted data from gastrocnemius samples of females (51 samples from distinct individuals aged between 24 and 70 years). The downloaded raw data from males were further processed with the customized workflow (see the next section). For females we used already pre-processed data ‘GTEx_Analysis_2017-06-05_v8_RNASeQCv1.1.9_gene_tpm.gct.gz’ available from the GTEx portal <https://www.gtexportal.org/home/datasets>.”

Supplementary Fig. 12. Changes in gene expression and pathway activities during *gastrocnemius* aging in female humans in comparison to other species. **a** Principal component analysis of transcript abundances during muscle aging in female humans. **b** PC1 coordinates for the RNA-Seq data set collected for female human *gastrocnemius* muscles. **c** Correlation between the standardized PC1 projections for individual genes (projection z-scores) for female and male humans. ‘hsa’ designates ‘Homo sapiens’ (human). ‘r’ indicates the value of the Pearson correlation coefficient. Black dashed line corresponds to the direction of the highest variance for the comparison, with the slope ‘s’ and intercept ‘i’. **d** Heatmap summarizing the enrichment of KEGG pathways among genes ranked by projection z-scores for mouse, rat and male and female humans, respectively. A pathway was included in the heatmap if it was significantly enriched in at least one organism with the significance threshold $FDR < 0.01$. Hierarchical clustering revealed pathways with a similar response during muscle aging in two or more species.

Minor comments:

1.7 (Page 1, line 14, Abstract): Please add “age related” so it reads “Age related loss of skeletal muscle...” for appropriate definition of sarcopenia.

We have updated the sentence to read (see l. 16-17, page 1):

“Sarcopenia, the age-related loss of skeletal muscle mass and function, affects 5-13% of individuals aged over 60 years.”

1.8 (Page 2, line 81, Results): How is healthy weight range defined in mice? Please consider adding literature or a definition.

We agree that given the lack of a clear definition for healthy weight range in mice, this statement is difficult to substantiate. We have now reworded the sentence to convey that at endpoint, mice were not obese (i.e. less than 30% body fat) and showed no obvious signs of cancer (see l. 94-95, page 3):

“At the time of dissection, body fat was below 26% for all mice, consistent with a lean phenotype²⁶, and showed no overt signs of tumors upon autopsy.”

1.9 (Page 3, line 89-91, Results): You have shown a great overview of % fat-loss, % reduction in tetanic force, % decrease in limb grip strength throughout this paragraph. Please consider including overall % loss of lean mass across hind-limb muscles in this paragraph, to help readers with overall comparisons.

To improve visualization of the extent of changes in whole-body and specific tissue mass, we now plot the body, fat, lean, muscle and organ mass at 28 months normalized to 8-18 months (i.e. adult baseline) in Fig. 1e, see the response to 1.2. Furthermore, we report percent differences relative to body mass and provide statistics. The data clearly show that, in mice, fat mass is preferentially lost. While muscle mass also outstrips body mass loss, it is counterbalanced by the increase in the mass of other lean tissues. We have added a sentence to this paragraph describing the range of hindlimb muscle loss (see l. 98-99, page 3):

“By 28 months, the muscle mass decreased by 19.7% in the tibialis anterior and 29.6% in the quadriceps, compared to 8-18 months-old mice.”

1.10 (Page 3, line 91-92, Results): I might misunderstand this figure. As I understand it, it shows loss of muscle mass relative to body weight at 28 months, normalized to 8-18 months. This highlights the predominant focus of muscle wasting on hind-limbs, this can be emphasized here and the overall conclusion (Page 3, line 106-109, Results). Given the decrease in lower skeletal muscle mass in humans associated with ageing this should be of great interest for the readership (For example see <https://doi.org/10.1152/jappl.2000.89.1.81> - Skeletal muscle mass and distribution in 468 men and women aged 18-88 yr).

Have you assessed whether loss of muscle mass outstrips loss of body mass overall? E.g. fat mass, liver, heart and epi. fat in the same graph and % loss of mass relative to body weight – I’d presume

this would even further strengthen the point that lean mass decreases most across hind-limbs in mouse, drawing a parallel to the human.

We found it a good suggestion to make it clearer how the loss of muscle mass compares to changes in the mass of other organs, tissues and body mass and rearranged Fig. 1e, see also responses to 1.2 and 1.9.

To strengthen the parallel between humans and rodents for muscle aging, we have added the following statement in the Discussion section (see l. 491-493, p. 12):

“In humans⁸⁹, rats¹² and mice muscle mass and strength are progressively lost with age. Our results demonstrate that rodent models also recapitulate the molecular changes observed in human sarcopenia.”

1.11 (Page 4, line 128-129, Results): For more clarity I recommend the authors make the following change:

“and young animals we observed changes in the expression of numerous genes (Supplementary Fig. 1), however, the vast majority of changes were smaller than 2-fold (Fig. 2g-h).”

We thank the reviewers for the suggestion, which we have implemented in l. 148-150, page 4.

1.12 Figure 1 legend

(Page 25, line 882) a > (a) (as for b, c, d, e & f in figure legend)

We thank the reviewers for pointing out this inconsistency. To avoid confusions with the style of panel labeling and to be consistent with the journal style, we omitted all parentheses for panel labels in figure captions. For highlighting panel labels we made them bold.

1.13 Figure 1 b: Lean mass > centered

We have corrected this as well (Fig. 1c).

1.14 Figure 1e: For more clarity I recommend the authors use the following labelling of Figure 1e.

“Loss of muscle mass relative to body weight at 28 mos (normalised to 8-18 mos, %)“

With the presumption that m was meant to abbreviate months, use the abbreviation mos.

In response to this and other comments, Fig. 1e has now been reworked, including the axis label (see the response to 1.2).

1.15 Figure 1 g-j: I recommend the authors add EDL to vertical labelling of figures, so it’s clear within the figure that these recordings have been undertaken on EDL alone.

We thank the reviewer for this suggestion. We included EDL in labels of y-axes of updated Fig. 1g-i (see the response to 1.2).

1.16 Figure 2 legend

(Page 25, line 896) a-c > (a-c) (as for d-f, g-i in figure legend)

See the response to 1.12.

1.17 Figure 3 legend

(Page 25, line 908) a > (a) (as for b-c in figure legend)

See the response to 1.12.

1.18 (Page 25, line 911) p-value0.05 change to p-value \geq 0.05

We thank the reviewers for pointing out the missing sign. It is now fixed (see l. 1189, page 28).

1.19 Figure 4 legend

(Page 25, line 917) a-c > (a-c) (as for d in figure legend)

See the response to 1.12.

1.20 (Page 25, line 919) I think the authors should show that ‘mmu’, ‘rno’ and ‘hsa’ abbreviations stand for = mus musculus, rattus norvegicus and homo sapiens.

We included the explanation of abbreviations in the figure caption (see l. 1156-1157, page 28):

“‘mmu’, ‘rno’ and ‘hsa’ designate ‘Mus musculus’ (mouse), ‘Rattus norvegicus’ (rat) and ‘Homo sapiens’ (human), respectively.”

1.21 Figure 5 legend

(Page 26, line 928) a > (a) (as for b in figure legend)

See the response to 1.12.

1.22 Figure 5 a legend

Please consider providing information to help readers interpret Figure 5a, currently this reads more like just the methodology used.

We rewrote the caption of Fig. 5a to point out that we depicted slopes of gene expression changes (see l. 1167-1174, page 28):

“... **a** Slopes of changes in the expression of genes from KEGG pathways that were significantly enriched for all considered species in GSEA ($FDR < 0.1$). The mean expression in replicates of the same age for each gene from the leading edge was calculated (see further description in the text). For humans, the mean expression was calculated for replicates from the age groups 20-29, 30-39, 40-49, 50-59, 60-69 and 70-79 years. The slopes defined by mean gene expression changes in neighboring time points (or age groups) were used to calculate median values across genes from the leading edge, which are visualized (Supplementary Fig. 7).”

1.23 Figure 6 legend

(Page 26, line 935) a > (a) (as for b, c-d in figure legend)

See the response to 1.12.

1.24 Figure 6, Supplementary Figure 8 and 9 – second column:

I might have missed this, but between which groups has the students t-test been performed and why? With multiple age groups I would expect to see comparison of the youngest with each of the other groups as undertaken previously.

The reviewers are correct, the two-sided Student's t-test for motif activities was performed selectively. To avoid confusion, we have now two-sided performed Student's t-test for motif activities of all ages/age groups in comparison to the youngest one, for all organisms. Test results are shown in Fig. 6c-e and Supplementary Fig. 8 and 9. We also included the interpretation of test results in figure captions (see l. 1186-1189, page 28):

“*, ** and *** denote a significant difference based on two-sided Student's t-test between the youngest age/age group and all other ages/age groups with $p\text{-value} < 0.05$, $p\text{-value} < 0.01$ and $p\text{-value} < 0.001$, respectively; 'n.s.' - not significant ($p\text{-value} \geq 0.05$).”

1.25 Supplementary Figures:

A light grey border appears around the rat and mouse schematics when using the zoom in/out function in supplementary figures. See screenshot attached, you might want to reformat the figures.

We thank the reviewers for pointing this out. The issue is now fixed.

1.26 Supplementary Figure 6:

“... (blue vector) on PC1 (black vector)” you mean “... (blue vector) on PC1 (blue vector)” (PC1 is blue)

To improve clarity, we introduced a new color scheme and added labels of the representative gene vector and its projection on the PC1 in Supplementary Fig. 6a (see below). The new color scheme is described in the figure caption.

Supplementary Fig. 6. Calculating projections of the gene expression on PC1. **a** Visualization of gene expression in the sample space. Each dot corresponds to a gene. Gene coordinates correspond to expression levels in individual samples, after subtraction of the mean across samples. Black

vectors designate PC1 and PC2 of the data set. Each gene can be associated with a vector starting from the origin and aiming at the point corresponding to the gene coordinates in the sample space (blue vector). Red dashed line indicates the projection of a representative gene vector (blue vector) on PC1 (black vector). **b** Distribution of projection values of gene expression vectors on the corresponding PC1 across species. **c** Distribution of projection z-scores across species.

Signed: Tom Gillingwater / Ines Boehm

Reviewer #2 (Remarks to the Author):

In the manuscript by Börsch, Ham et al., the authors employ a comprehensive phenotypic and bioinformatic approach to identify molecular signatures of age-associated loss of skeletal muscle mass and function i.e. sarcopenia. By utilizing three different model systems (human, rats and mice), they sought to identify common transcriptional signatures that are associated with muscle aging. In addition, the authors make use of the temporal information in the data sets in order to attempt to uncover novel pathways that could be driving the loss in muscle function with age. Their computational pipeline leads to a number of molecular pathways that are conserved across species during aging, most notably the loss of mitochondrial function and increased inflammation. Lastly, their datasets provide insights into which rodent model should be used depending on what biological process/pathway you are studying, particularly if wanting to translate the findings into a human population.

Overall, the manuscript is well-written and the computational approach is quite sophisticated, with their results and ‘SarcoAtlas’ undoubtedly being a valuable resource for researchers studying muscle biology during aging. My specific comments are listed below:

Major Comment

2.1 - The authors did a very nice job of utilizing the temporal resolution in their data sets to identify potential drivers of muscle aging; however, the results from this computational approach would have benefited greatly from a final validation step showing differences in the abundance/activity of the TFs that are proposed to contribute to the differences in the transcriptional signature. This would add strength and confidence that the computational approach was able to uncover novel targets involved in skeletal muscle aging. This could be done just in the rodent models, since tissue samples are easier to obtain.

We agree with the reviewer that an experimental validation of computationally predicted changes in expression/abundance/activity of transcription factors (TFs) during muscle aging would strengthen our conclusions. With the limited material that we had from our mouse aging system, we were able to analyze the abundance of $ERR\alpha$, $PPAR\alpha$ and $YY1$ proteins, and found that it is significantly lower in the *gastrocnemius* muscle of aged compared to young mice (Fig. 6f, g, see below). Furthermore, for the $ERR\alpha$ TF, which was significantly downregulated during muscle aging at both transcript and protein levels, we visualized its co-localization with nuclei of young and old *tibialis anterior* mouse muscles. We also showed that the fraction of $ERR\alpha$ -positive nuclei, i.e. nuclei that were at least 50% covered by the $ERR\alpha$ stain, was much higher in the young compared to the old muscle, indicating the higher activity of the TF $ERR\alpha$ in muscles of young

mice. We carried out the microscopy analysis in *tibialis anterior* and not in *gastrocnemius* due to the availability of the biological material suitable for this kind of analysis. These experimental measurements support the results of our computational analysis and confirm their validity. Finally, we also included a visualization of our RNA-Seq data sets, which demonstrates that the expression of *ESRRA* is downregulated during aging in all species and the expression of *PPARA* is decreased in rats and humans (Supplementary Fig. 10, see below). These findings were summarized in the manuscript (see l. 296-312, page 8):

“To partially validate our predictions, we checked the RNA levels of TFs commonly regulated across species in our data sets, as well as the protein abundance and localization of a select set of TFs in mouse muscle tissue. RNA expression of ESRRA was significantly downregulated during muscle aging in all species, while PPARA transcripts were significantly decreased in rat and human aging (Supplementary Fig. 10). RNA of genes encoding other TFs with motifs exhibiting age-related changes were either unchanged or too lowly expressed for statistical testing. We further analyzed protein expression of the TFs ERRA, PPARα and YY1, whose activity was predicted to decrease with age in all species, by western blotting (Fig. 6f). In line with our predictions, the relative abundance of ERRA, PPARα and YY1 was lower in 28 than 8 months-old mouse gastrocnemius muscle (Fig. 6f, g). Since ERRA exerts its activity in the nucleus, we further quantified the prevalence of nuclei strongly positive for ERRA staining (ERRA+; >50% of nuclear area with positive ERRA staining) in 8 and 28 months-old tibialis anterior muscle cross sections (Fig. 6h). ERRA+ nuclei were prevalent in 8 months-old muscle, displaying ~80 ERRA+ nuclei per 100 muscle fibers. Aging drastically reduced the prevalence of ERRA+ nuclei, with 28 months-old muscle showing 4-fold less ERRA+ nuclei than 8 months-old mouse muscle (Fig. 6i). Together, these data support our computational predictions of TF activity in aging muscles.”

Fig. 6. ISMARA-inferred³⁹ activity of transcription factors (TFs) and miRNAs during muscle aging. **a** Venn diagram of motifs associated with TFs decreasing the activity during muscle aging. **b** Venn diagram of motifs associated with TFs and miRNAs whose targets were upregulated during muscle aging. Names of TFs and miRNAs are color-coded with respect to species: orange- mouse, brown- rat, black- human. **c-e** Activity of motifs associated with TFs Esrrb/Esrra, Esrrb/Essra and ESRR/ESR2 commonly regulated in mouse, rat and human, respectively. The 1st column depicts the normalized expression (z-scores of mean log₂(TPMs)) of the top 300 target genes. The mean value per age (or age group) across genes is indicated by the black (reference) line. Gene expression time course lines were colored by the distance from the reference line: red- close to the reference line, blue- far from the reference line. The 2nd column depicts the activity of the corresponding TFs predicted by ISMARA. *, ** and *** denote a significant difference based on two-sided Student's t-test between the youngest age/age group and all other ages/age groups with p-value<0.05, p-value<0.01 and p-value<0.001, respectively; 'n.s.'- not significant (p-value≥0.05). The 3rd column shows the 10 most enriched GO terms for the top 300 target genes of the TFs, i.e. genes depicted in the 1st column. GO analysis was performed in DAVID⁹⁷. Red dashed lines indicate the significance threshold (p-value<0.01). The numbers next to the bars denote how many genes were attributed to an enriched GO term. **f** Representative western blot analysis of the abundance of TFs ERR α , YY1 and PPAR α in the *gastrocnemius* muscle (tissue lysate) of 8 and 28 months-old mice, respectively. **g** Quantification of western blots showing the relative abundance of TFs ERR α , YY1 and PPAR α normalized to the nuclear protein histone H3, respectively. * denotes a significant difference based on two-sided Mann-Whitney U test between 8 and 28 months-old mice with p-value<0.05. **h** Representative images of *tibialis anterior* cross sections of 8 and 28 months-old mice with magnification stained for ERR α (red), Laminin α 2 (white), and DAPI (blue). **i** Quantification of the percentage of ERR α -positive nuclei in *tibialis anterior* fibers of 8 and 28 months-old mice, respectively. *** denotes a significant difference based on two-sided Student's t-test between 8 and 28 months-old mice with p-value<0.001.

Supplementary Fig. 10. Expression of genes encoding TFs commonly regulated during *gastrocnemius* muscle aging in mouse, rat and human. *, ** and *** denote a significant difference based on differential expression analysis between the youngest age/age group and all other ages/age groups of FDR<0.05, FDR<0.01 and FDR<0.001, respectively; 'n.s.'- not significant (FDR≥0.05). Lowly expressed genes were excluded from the statistical analysis and their expression plots are shaded.

Minor Comments

2.2 - I'm curious as to why the significant drop in muscle mass is not associated with a significant drop in total lean mass, particularly since the authors point out that skeletal muscle can contribute to up to 50% of body weight. Are other organs making up the difference?

We thank the reviewer for raising this point. We have adapted Fig. 1 (see the response to 1.2) to show more clearly that while muscle loss outstrips body mass loss, other lean tissue organs (e.g. liver and heart) either maintain or increase their mass, balancing out muscle loss. Importantly, although not directly measured in this study, increased seminal vesicle mass was a large contributor to the increase in non-skeletal muscle mass. We previously reported (Ham and Börsch, Nature Communications, 2020, <https://doi.org/10.1038/s41467-020-18140-1>, Supplementary Fig. S2b) a five-fold age-related increase in seminal vesicle mass from 435mg in 10 months-old mice to 2223 mg in 30 months-old mice. We also kindly ask the reviewer to refer to the response to 1.9. The relevant text has been updated accordingly (see l. 102-108, page 3):

“Analysis of internal organs showed a maintenance (liver) or increase (heart) of other lean-tissue organs (Fig. 1e), explaining the overall maintenance of whole-body lean mass. Indeed, seminal vesicles, although not measured in this study, were notably larger in older mice, as previously observed²⁷. In contrast, and consistent with MRI measures of body composition, epididymal fat depositions were measurably lower in 22-28 months-old mice than 8-18 months-old mice, reaching a body mass normalized reduction of $81.6 \pm 2.8\%$ at 28 months.”

2.3 - The authors mention that they used the gastrocnemius from the rodents because there was corresponding data in the human population. Is there any phenotypic data for the human population? Could this possibly explain why there is such a small number of differentially expressed genes in the human dataset? It's possible that the muscle health of these subjects was relatively normal, and that was why the transcriptional profile did not differ that much from the young counterparts.

We thank the reviewer for these questions. Concerning the availability of phenotypic measurements for the human population, we kindly ask the reviewer to refer to the response to 1.3, where this issue was addressed. Here, we address the question about the number of differentially expressed (DE) genes in the human data set. Several factors could explain the relatively small number of genes with significantly changed expression between 'young' and 'old' groups of humans. First, as the reviewer correctly pointed out, RNA-Seq data sets for rodents, particularly mice, covered a longer lifespan than the RNA-Seq data set for humans. We commented on this in the Discussion section (see l. 339-348, page 9):

“Mapping molecular changes occurring during muscle aging between species is also not straightforward. Orthologous genes may change expression with different dynamics in different species (Fig. 5a). Moreover, the lifespan represented in various data sets may differ for different species, which may result in different numbers of differentially expressed genes between 'young' and 'old' individuals. Indeed, the age range represented in the mouse, rat and human data was quite different, as 24 months of age in rats is thought to correspond to 60 years of age in humans⁵⁰,

and 20 months of age in mice⁵¹. Thus, our mouse data corresponds to a much higher age compared to the data for rats and humans. This could explain the larger number of genes found as differentially expressed between ‘young’ and ‘old’ mice compared to humans (Supplementary Fig. 1).”

Second, the human data set is highly heterogeneous and much more complex compared to the data sets obtained for rodents grown in laboratory conditions. The heterogeneity might decrease the power of statistical methods for finding DE genes. We included this hypothesis in the Discussion section (see l. 359-367, page 9):

“Moreover, the human population is particularly heterogeneous and human data sets are much more complex in comparison to data sets obtained for rodents grown in laboratory conditions. Indeed, the median value of the coefficient of variation calculated for the gene expression in replicates belonging to the same age/age group and reflecting the dispersion rate in the gene expression in replicates of the same age/age group was much bigger for humans than for rodents varying for different ages/age groups between 26% and 34% in rodents and between 34% and 60% in humans (Fig. 2d-f, thick blue lines within violin plots). This heterogeneity may decrease the power of statistical methods for finding genes differentially expressed between ‘young’ and ‘old’ groups.”

2.4 - Similarly, the authors should comment on how representative the respective age groups of the model systems are to one another. This fact was touched on a bit, but it should be mentioned that the human age group is still relatively “young” compared to the mice, who are extremely old. Without phenotypic data, it’s hard to know whether the human population is sarcopenic, thus complicating the interpretation.

The reviewer is correct that the age span of the human samples does not reach the length of the mouse samples (see a more precise estimation in the response to the previous comment). Although we are limited in phenotypic measurements for the human population, we observed similar molecular changes occurring during muscle aging in rodents, where we directly related phenotypic changes to molecular changes represented by PC1. Therefore, we analyzed the human data similarly to the data from rodent species, i.e. from the perspective of PC1. We introduced this assumption in the manuscript (see l. 191-195, page 5):

“... the distribution of samples on PC1 followed a similar pattern in humans as in other species, with higher between-subject variance in PC1 coordinates for individuals 46-70 years of age compared to those 45 years or younger (Supplementary Fig. 5b). Therefore, we analyzed both human and rodent data from the perspective of PC1.”

We agree with the reviewer that additional studies, perhaps including more direct estimates of muscle state, are needed to verify our hypothesis that PC1 reflects this state in humans, as it does in rodents. Here we have extracted the most we could of the currently available data (see l. 195-197, page 5):

“To verify the assumption that PCI of the human data reflects muscle health, further studies of human sarcopenia combining thorough phenotyping with molecular profiling of muscle samples are needed.”

Reviewer #3 (Remarks to the Author):

Brief summary of the manuscript

Börsch et al. analyzed and compared skeletal muscle phenotypic and molecular parameters and their age-projections between rodent animal models and humans with the goal to evaluate the usefulness of animal models to recapitulate human skeletal muscle aging. The authors based their conclusions on skeletal muscle and whole body phenotypic data (collected for mice in this study; body mass, muscle mass and function), and molecular data (all three species, RNAseq from mouse (this study), rat (this and previously published study), and human (using a public database)) over a spread of ages. Gene and pathway correlations between species were analyzed with statistical methods and models. The main finding of this study is that rodents indeed seem to recapitulate human muscle aging well, but the authors qualified that one or the other rodent species better represents specific aspects of human muscle aging (such as inflammation or metabolism). Further, 1) human muscle aging seems to be recapitulated by rodent models on the pathway rather than the individual gene level, and 2) the timing of muscle aging, reflected in changes in pathway regulation, differs between rodents and humans.

Overall, this study represents an interesting body of analytical and computational work that transforms individual gene expression analysis across species into a correlation analysis of molecular pathways. The authors created and utilized existing large data sets and comprehensively assessed skeletal muscle aging in this cross sectional study. Interestingly, the authors found no key driver of skeletal muscle aging on the individual gene level, no obvious strong correlation between “skeletal muscle health” and chronological age, and only a handful of differentially expressed individual genes overlapping between the species, diminishing enthusiasm in limited gene expression preclinical studies for the evaluation of human aging. Molecular data seem to correlate better than chronological age with phenotypic data, with pathways correlating better between the species and with aging than individual gene expression data. The authors further present an array of differentially expressed molecular pathways shared between at least two species as future target pathways for rodent studies recapitulating human skeletal muscle aging. What is furthermore interesting is the report of age-related changes in transcription factor activities.

The findings gained from these analyses could have significant implications for the design and interpretation of future studies investigating muscle aging in animal models as well as in humans. In addition, the authors’ data and main conclusions have built a basis for future investigations into the regulation of muscle aging and potential interventional targets at the level of transcriptional regulators. It is therefore of interest to the wider field of aging biology on both the preclinical and the clinical level.

The authors define in the methods section the way the data are calculated; the datasets are accessible; and for a researcher well versed in the respective computational methods, the data

should be reproducible. Nevertheless, it is evident that the authors speak to a subset of aging researchers with more extensive knowledge of statistical analyses and modeling. The authors explained (at least for the most part) how they derived the PCA data and subsequent computations, using “toy data sets”.

However, although PCA of large data sets such as RNAseq data is not uncommon, the results and additional data derivations such as coefficient of variance analysis (Fig 2d-f) presented here are not always sufficiently explained for the wider target audience that could and would want to take advantage of the authors’ findings and conclusions. It is not completely obvious why specific analyses were chosen, how the resulting data were derived, and what can be learned from these derivations.

The reader would benefit from more explanation (without creating a statistics textbook). This shortcoming prevents this reviewer and possibly a wider audience from fully evaluating the validity of the results, appreciating the conclusions drawn from them, and implementing the presented insights into future study design and analyses.

Specifically:

3.1 Figure 2a-c: The PCA of the molecular RNAseq data revealed that PC1 in the three species explains 37 to 18% of the variance between samples within each species and PC2 8 to 14% (Fig 2a-c). However, it is not elaborated on which major genes comprise PC1 and PC2, and what their respective weights (or loads) are. Is there any potential information to be gained from disclosing the genes that provide most weight to the PC1 (and 2) and for the nature of the genes’ loading scores? If so, please add, if not, please explain why not.

We thank the reviewer for emphasizing the relevance of our study and appreciate the questions, which helped us to improve the presentation of the material and make it accessible to a wider audience. In particular, we realized that differences in the vocabulary related to the principal component analysis between the literature and our manuscript may be confusing. To explain the basic principle of the analysis, we introduced a toy example and used what we think is a more intuitive terminology in the context of the example (Supplementary Fig. 6a). Our ‘projection values’ have the same meaning as the ‘weights’/‘loads’/‘loadings’ from the literature. Principal components create a new basis for genes located in sample space (Supplementary Fig. 6a). Thus,

the expression of a gene g_1 in samples s_1, s_2, \dots, s_n can be represented as the linear combination of principal components PC1, PC2, ..., PCn:

$$g_1 = l_{11}PC1 + l_{12}PC2 + \dots + l_{1n}PCn,$$

where l_{11} is the coordinate of g_1 on PC1 defined by the projection of g_1 on PC1, l_{12} is the coordinate of g_1 on PC2 defined by the projection of g_1 on PC2 and so on. Coordinates $l_{11}, l_{12}, \dots, l_{1n}$ of the gene g_1 in the space of principal components are also called ‘weights’/‘loads’/‘loadings’ in the literature. We added this clarification in the Methods section (see l. 707-716, page 18).

Indeed, as the reviewer suggests, we think it is useful to study genes with high and low projections on PCs. According to our analysis, genes with positive projections on PC1 increased whereas that

of genes with negative projections on PC1 decreased their expression during sarcopenic progression. For each organism we ordered genes by their projections on PC1 and subjected them to gene set enrichment analysis (GSEA) to reveal KEGG pathways associated with loss of *gastrocnemius* functionality during aging across species (Fig. 4d, e). We have further provided a user-friendly web-application 'SarcoAtlas' (<https://sarcoatlas.scicore.unibas.ch/>) that allows interested readers to perform the PC analysis on their own, to obtain genes that align best to each PC, including the z-score of their projection on the PC and the correlation to the PC.

3.2 Figure 2d-f: The next piece of information given is plotted in Figure 2d-f, which aims to show the scatter of gene expressions along PC1 and 2, and potential age-clustering (or the lack thereof). The authors claim that the variability increases with age (Fig. 2d-f); however, without more explanation of these plots (Fig. 2d-f) and their construction, and the description in the results section (line 127-127) it remains rather obscure (what am I looking at, where is the higher variation in older groups compared to younger, what do the black lines/outlines mean?). Understanding these data will help appreciating the conclusion, that chronological age does not necessarily correlate with gene expression changes.

We appreciate the reviewer's comment, which we addressed by including a detailed explanation of how we calculated the coefficient of variation and what it indicates. Specifically, the CV is a measure of dispersion of a distribution, which in our case (Fig. 2d-f) shows that gene expression levels vary more among individuals of the same age, as the age increases. We introduced a new subsection within the Methods (see l. 652-665, p. 16-17):

“Calculating the coefficient of variation of gene expression levels per age/age group”

The coefficient of variation (CV) is a measure of dispersion of a distribution, often used to assess the gene expression heterogeneity among replicate samples. We have computed the CVs for our data sets to assess the heterogeneity in gene expression as a function of age. For each organism, age/age group and gene we calculated the mean value of the expression (in TPM units) and standard deviation of the expression across replicates. We then computed the coefficient of variation (CV) as the ratio of the standard deviation to the mean. Then, for each organism and each age/age group we plotted the distribution of CVs across all genes in the form of violin plots (Fig. 2d-f). We took the median value of CVs for the youngest age/age group as a baseline (thin blue line drawn across all ages/age groups). Median values of CVs for other ages/age groups (thick blue lines within violin plots) are mostly located over the baseline, especially for older ages/age groups, indicating higher heterogeneity across replicates for these ages/age groups in comparison to the youngest one.”

We also revised the legend of Fig. 2d-f, where we also included the interpretation of CV for our analysis (see l. 1127-1135, page 27):

*“... **d-f** Distribution of the coefficients of variation (CVs) of individual genes per age/age group for mice in **d**, rats in **e** and humans in **f**. The higher the CV, the more variable the expression of the gene across replicates. Thin blue lines are baselines indicating the median coefficient of variation*

for the youngest age/age group. Median values of CVs for other ages/age groups (thick blue lines within violin plots) are mostly located over the baseline, especially for older ages/age groups, indicating higher heterogeneity across replicates for these ages/age groups in comparison to the youngest one. Limits for y-axis were set to include both the 25th and 75th percentiles of data points and up to 1.5 times the interquartile range in both directions from percentiles (red).”

3.3 Figure 3: The authors went on to show that PC1 correlates with phenotypic data better than with age. This is not entirely obvious from Figure 3a, since the correlation coefficients between PC1 and age versus PC1 and muscle mass do not seem to differ much (both ~0.7-0.75). However, this claim seems more apparent in Fig. 3b. Please clarify this discrepancy between Fig 3a and 3b? Are the graphs in the Supplementary Figure 2 and 3 (age vs PC1) the same data shown in Fig 3b?

We thank the reviewer for pointing out the visual effect, which may induce the reader to doubt the coherence of our analysis. The cause may have been the poor y-axis labeling of Fig. 3a. If we refine the grid of Fig. 3a, include additional grid labels and put equal limits for the y-axis on the positive and negative sides, the discrepancy disappears (see below). To make it even more apparent, we added two dotted horizontal lines representing the negative value of the correlation between PC1 and age for mouse and rat, respectively. This description was also added to the figure caption (see l. 1144-1145, page 27).

Fig. 3. Correlation between PC1 and phenotypic measurements in rodents. **a** Values of the Pearson correlation coefficient between PC1 and phenotypic measurements for mice (black) and rats (red) during aging. Black and red dotted horizontal lines are baselines representing the negative value of the correlation coefficient of PC1 with age for mouse and rat, respectively. Fisher's test¹⁰⁰ was used to compare Pearson correlation coefficients obtained for the same measure for mouse and rat, *p-value<0.05, **p-value<0.01, ***p-value<0.001, 'n.s.' -not significant (p-value≥0.05). **b-c** PC1 coordinates of mouse and rat RNA-Seq data sets, respectively, grouped by age and colored by the *gastrocnemius* mass. With dashed ellipses we marked two groups of 28 months-old mouse replicates ('G1' and 'G2') used for subsequent differential expression analysis. Abbreviations: BM- body mass, GAST- *gastrocnemius*, TA- *tibialis anterior*, EDL- *extensor digitorum longus*, SOL- *soleus*, QUAD- *quadriceps*, PLA- *plantaris*.

3.4 Figure 4: Figure 4 builds on projection values and their standardization by z-scores. For clarification (from which the reader would likely benefit as well): does the magnitude of the gene's projection vector onto PC1 correspond to this variable's load? Further, please add why normal distribution of the projection of gene vectors onto PC1 was expected (line 177)?

We kindly ask the reviewer to refer to the response to 3.1, where we explained the connection between projections of gene vectors on PC1 and values of 'weights'/'loads'/'loadings'. Here, we discuss why projections of gene vectors on PC1 are normally distributed. It has been empirically observed that normalized gene expression levels in TPM units follow log-normal distribution (Beal, J. Biochemical complexity drives log-normal variation in genetic expression. *Engineering Biology* vol. 1 55–60 (2017), <http://dx.doi.org/10.1049/enb.2017.0004>). Prior to principal component analysis, our data sets containing gene expression levels in TPM units were log2-transformed, i.e. brought to the normal distribution, and mean centered. Mean centering and projecting data on a vector (in our case it is PC1) are linear transformations preserving the distribution of the data. The explanation was added to the Methods section (see l. 702-706, page 18):

“Note that projections of gene vectors on PC1 are normally distributed for all considered species (Supplementary Fig. 6b). This stems from the fact that log2-transformed normalized gene expression levels in TPM units are close to normally distributed⁹⁶. Mean centering and projecting data on a vector (in our case it is PC1) are linear transformation methods, which preserve data distribution.”

3.5 Figure 6 (particularly 6 c-e and respective supplementary figures) is not well explained. Is any of the TF activities significantly different between ages or age-groups (only human age group 2029 is indicated to be significantly different from 30-39)? The third column in Figure 6 c-e needs to be clarified: enriched GO terms, meaning TF activity associated with those pathways decrease (as indicated in the other two columns)?

We kindly ask the reviewer to refer to the response to 1.24, where we describe the performed two-sided Student's t-tests for motif activities in our data sets predicted by ISMARA and added test results to Fig. 6c-e and Supplementary Fig. 8 and 9 and the interpretation of test results to the figure captions. We also included more clarification on how the 3rd column of Fig. 6c-e and

Supplementary Fig. 8 and 9 was composed in the caption of these figures (see l. 1189-1191, page 28) and in the Methods section (see l. 777-781, page 19):

“The 3rd column shows the 10 most enriched GO terms for the top 300 target genes of the TFs, i.e. genes depicted in the 1st column.”

“To characterize processes regulated by conserved TFs, we subjected their top 300 targets (depicted in the 1st column of Fig. 6c-e and Supplementary Fig. 8, 9) to the gene ontology (GO) analysis using Database for Annotation, Visualization and Integrated Discovery (DAVID)⁹⁷ through the R/Bioconductor package called ‘RDAVIDWebService’⁹⁸ (3rd column of Fig. 6c-e and Supplementary Fig. 8, 9).”

3.6 Line 362: was PGC1a also found downregulated in the three data sets?

We appreciate the reviewer’s interest in the expression of PGC-1a. The expression was plotted in Supplementary Fig. 11 (see below). It is significantly downregulated in rats starting from the age of 20 months and has a decreasing trend in humans. We also included a note about it in the main manuscript (see l. 444-448, page 11):

“In our data sets, the expression of the gene *PPARGC1A* encoding PGC-1a is significantly downregulated in rat gastrocnemius starting from the age 20 months and has a decreasing trend during muscle aging in humans (Supplementary Fig. 11). PGC-1a expression in mice remains unchanged in our RNA-Seq data set, but its decline in mouse skeletal muscles was previously reported in other studies⁸³.”

Supplementary Fig. 11. Expression of the gene encoding PGC-1a (*Ppargc1a* in rodents, *PPARGC1A* in human) during *gastrocnemius* muscle aging in mouse, rat and human. *** denotes a significant difference based on two-sided Student’s t-test between the youngest age group and all other age groups of p-value<0.001; ‘n.s.’- not significant (p-value≥0.05).

3.7 Line 312 and following: reference to the NIA Rapamycin study could be useful (PMID: 19587680).

We thank the reviewer for pointing out the study on the role of mTORC1 and its inhibitor, rapamycin, in aging. We indeed believe that mTORC1 is one of the central players in muscle aging. In fact, in a recent study, we demonstrated that the neuromuscular junction is a focal point of mTORC1 signaling in sarcopenia (Ham and Börsch, Nature Communications, 2020, <https://doi.org/10.1038/s41467-020-18140-1>), and that chronic inhibition of mTORC1 with

rapamycin has an overwhelmingly, but not entirely, positive effect on the mouse skeletal muscle during aging. We extended the discussion about the role of mTORC1 and rapamycin in aging and, in particular, muscle aging (see l. 382-386, page 10):

“Inhibition of mTORC1 with rapamycin provided in the food extended the mouse lifespan when administration started at 600 days of age⁵⁶, while in a recent study of the muscle we have found that chronic mTORC1 inhibition with rapamycin is overwhelmingly, though not entirely, positive for aging mouse skeletal muscle²⁵.”

REVIEWERS' COMMENTS:

Reviewer #1 (Remarks to the Author):

The authors have done a fantastic job of responding to the initial referees' comments, and have further improved what was already an excellent paper. There are no further comments/concerns to raise.

Reviewer #2 (Remarks to the Author):

To all the authors - very nice job responding the comments/concerns brought up by the reviewers. You addressed all of my concerns and the I believe that the additional TF validation data strengthens the manuscript nicely. This paper will be a great resource for anyone studying the progression of sarcopenia. Very nice work!

Reviewer #3 (Remarks to the Author):

The authors have addressed my comments to my satisfaction and overall improved the manuscript. I recommend publication of this manuscript. - Thank you!

We thank reviewers for assessing our manuscript one more time and providing positive feedback on our work.

REVIEWERS' COMMENTS:

Reviewer #1 (Remarks to the Author):

The authors have done a fantastic job of responding to the initial referees' comments, and have further improved what was already an excellent paper. There are no further comments/concerns to raise.

Reviewer #2 (Remarks to the Author):

To all the authors - very nice job responding the comments/concerns brought up by the reviewers. You addressed all of my concerns and the I believe that the additional TF validation data strengthens the manuscript nicely. This paper will be a great resource for anyone studying the progression of sarcopenia. Very nice work!

Reviewer #3 (Remarks to the Author):

The authors have addressed my comments to my satisfaction and overall improved the manuscript. I recommend publication of this manuscript. - Thank you!